# Dataset Condensation with Color Compensation

**Huyu Wu**[1]    **Duo Su**[2*]    **Junjie Hou**[3]    **Guang Li**[4*]
[1] *University of Chinese Academy of Sciences*    [2] *Tsinghua University*
[3] *Hong Kong University of Science and Technology*    [4] *Hokkaido University*
[*] *Corresponding Authors:* `suduo@mail.tsinghua.edu.cn, guang@lmd.ist.hokudai.ac.jp`

**Reviewed on OpenReview:** *`https://openreview.net/forum?id=hIdwvIOiJt`*

## Abstract

Dataset condensation always faces a constitutive trade-off: balancing performance and fidelity under extreme compression. Existing methods struggle with two bottlenecks: image-level selection methods (Coreset Selection, Dataset Quantization) suffer from inefficiency in condensation, while pixel-level optimization (Dataset Distillation) introduces semantic distortion due to over-parameterization. With empirical observations, we find that a critical problem in dataset condensation is the oversight of color's dual role as an information carrier and a basic semantic representation unit. We argue that improving the colorfulness of condensed images is beneficial for representation learning. Motivated by this, we propose **DC3**: a **D**ataset **C**ondensation framework with **C**olor **C**ompensation. After a calibrated selection strategy, DC3 utilizes the latent diffusion model to enhance the color diversity of an image rather than creating a brand-new one. Extensive experiments demonstrate the superior performance and generalization of DC3 that outperforms SOTA methods across multiple benchmarks. To the best of our knowledge, besides focusing on downstream tasks, DC3 is the first research to fine-tune pre-trained diffusion models with condensed datasets. The Frechet Inception Distance (FID) and Inception Score (IS) results prove that training networks with our high-quality datasets is feasible without model collapse or other degradation issues. Code and generated data are available at `https://github.com/528why/Dataset-Condensation-with-Color-Compensation`.

## 1 Introduction

In data-centric artificial intelligence, the dataset serves as a vehicle of knowledge whose information density and representation capacity determine the cognitive boundaries of a model. Information theory reveals a fundamental contradiction: There exists a coexistence of sparse task-relevant information and redundant task-irrelevant noise within raw data (Shannon, 1948; Cover, 1999; MacKay, 2003). Inspired by this contradiction, researchers propose dataset condensation, which focuses on creating a small dataset from the original large-scale data to reduce training costs while preserving comparable performance (Geng et al., 2023; Lei & Tao, 2023a; Yu et al., 2023a). The condensed dataset not only saves storage space but also accelerates I/O speed.

**D**ataset **C**ondensation (DC) includes **C**oreset **S**election (CS) (Guo et al., 2022; Sinha et al., 2020; Rosman et al., 2014; Chai et al., 2023), **D**ataset **Q**uantization (DQ) (Zhou et al., 2023; Zhao et al., 2024), and **D**ataset **D**istillation (DD) (Liu et al., 2022; Du et al., 2023a; Zheng et al., 2024). As illustrated

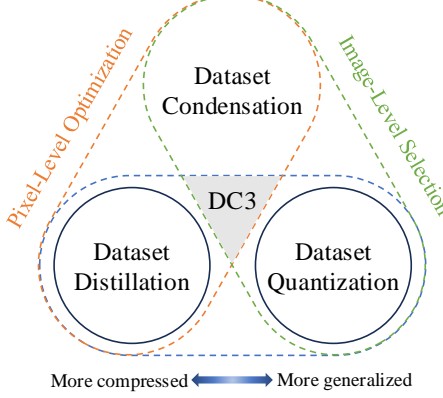

Figure 1: **Dataset Condensation Triangle.** Coreset Selection and Dataset Quantization focus on image-level selection, while Dataset Distillation optimizes pixels for better condensation. An ideal DC method should balance information compression and semantic preservation.

in fig. 1, we elucidate different DC methods as **DC Triangle**. The primary distinction of these approaches is that CS and DQ focus on image-level selection, thereby exhibiting better cross-architecture generalization. DD implements pixel-level optimization that results in pronounced condensation performance (Cazenavette et al., 2022). Despite reducing dataset scale and enhancing computational efficiency, the synthetic image generated by DD lacks sufficient and consistent semantic information (Su et al., 2024). The compression capability of the matching-based distillation strategy is achieved at the expense of its semantic realness. The optimal synthetic image should neither be too realistic nor too expressive (Cazenavette et al., 2023; Du et al., 2023b). Motivated by this observation, we seek to propose a method that not only ensures condensation capability like DD but also guarantees the lossless preservation of semantic information like CS and DQ.

The essence of dataset condensation lies in exploring how the models extract information from a dataset and what the models actually "see"? Color plays a crucial role in the first principle of human visual perception, forming the basis of how we understand the world (Jameson et al., 2020; Boynton, 1990). Similarly, color has a dual identity in machine visual compression: it is both the physical signal that needs optimizing (Prativadibhayankaram et al., 2023) and the basic unit that maintains the semantic relevance of an image (Van Leeuwen, 2011). Such a dual nature makes color a main "battlefield" in dataset condensation, which previous methods have overlooked.

As depicted in figs. 2a to 2c, the RGB channels of the distilled datasets follow different distributions compared to the original datasets (especially for the matching surrogate objective methods (Zhao & Bilen, 2023; 2021a; Sajedi et al., 2023). Their Kernel Density Estimation (KDE) curves all tend to have a uniform distribution. Such ***Color Homogenization*** phenomenon could hinder the model in understanding the data representations, thus degrading its generalization. Unlike DD, quantized datasets are derived through a quantization process from the original data, notably preserving precise color distribution and semantic integrity, whereas lacking the information compression capability inherent to DD. Motivated by these observations, we seek to study: "*Can we avoid **Color Homogenization**, but attain an informative dataset by directly using the image-level selection?*"

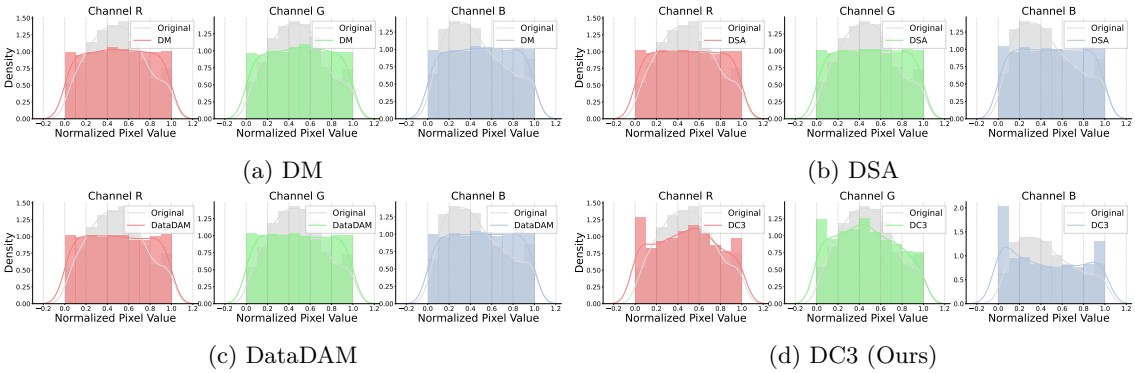

Figure 2: **The KDE curves of the normalized RGB pixel value across different DC methods.** Previous methods exhibit a ***Color Homogenization*** phenomenon in contrast to DC3.

Recent advancements in generative modeling have established diffusion models as a dominant approach for text-to-image and image-to-image generation. Building upon these successes, pioneering works have adapted diffusion models to dataset condensation, leveraging their high-fidelity generation capability to preserve essential characteristics within compressed datasets (Su et al., 2024; Cazenavette et al., 2023; Gu et al., 2024; Sajedi et al., 2024). However, diffusion models exhibit limited robustness under sparse or ambiguous textual conditions, generating images with semantic inconsistency or hallucination that deviate from real-world distributions.

In this paper, we propose a **D**ataset **C**ondensation framework with **C**olor **C**ompensation (**DC3**) to enhance the quality of condensed datasets. DC3 adds color-related hues to images via a pre-trained diffusion model. As depicted in fig. 2d, the KDE curve is closer to the real dataset than other methods, which is conducive to enhancing the representation capability. Next, through analyzing the relationship between submodular gain

and distribution of samples, DC3 reuses the submodular gain to select samples, while the bin generation stage is handed over to K-Means clustering. Consequently, the whole image-level selection process has become more controllable and efficient. DC3 is also a training-free method, whose processing speed depends on the sampler and diffusion networks. The pivotal contributions are summarized as follows:

- According to the dataset condensation triangle, we propose DC3 that utilizes the compression ability of dataset distillation along with the generalization ability of dataset quantization. It is a universal method that is adaptive to datasets of various scales and resolutions.

- Due to ***Color Homogenization*** inherent in pixel-level optimization methods, we employ the clustering-based quantization method to get rid of this issue and propose the diffusion-based ***Color Compensation*** method to enhance the information diversity of condensed images.

- In addition to the classification task, DC3 also dives into improving the fine-tuning performance of large vision models (LVMs). The FIDs on the fine-tuned stable diffusion and DiT demonstrate that the information is compressed and preserved by DC3.

- The experimental results, especially on the hard-to-classify datasets, demonstrate that DC3 achieves the superior performance of dataset condensation and enhances image colorfulness.

## 2 Related Work

### 2.1 Dataset Distillation

Dataset Distillation (DD) (Wang et al., 2018; Li et al., 2022; Lei & Tao, 2023b; Yu et al., 2023b) methods are categorized into matching-based and generation-based approaches. Matching-based methods can also be categorized into three paradigms: performance matching, parameter matching, and distribution matching. Performance matching focuses on optimizing the synthetic dataset to ensure that models trained on it perform similarly to those trained on the original dataset (Nguyen et al., 2021a;c). Parameter matching, which includes single-step (e.g., gradient matching (Zhao & Bilen, 2021b;a)) and multi-step (e.g., trajectory matching (Cazenavette et al., 2022; Guo et al., 2024)) approaches, aims to align the model parameters or their learning trajectories during training on synthetic and real data. Distribution matching emphasizes narrowing the distribution gap by ensuring that synthetic data approximates the feature distribution of the original data (Zhao & Bilen, 2023; Deng et al., 2024).

Generation-based methods have recently gained traction by leveraging modern generative models to bypass the limitations of iterative sample-wise optimization (Zhao & Bilen, 2022; Cazenavette et al., 2023). $D^4M$ (Su et al., 2024) uses latent diffusion models to synthesize data. It maintains consistency between real and synthetic image spaces and incorporates label prototypes. Minimax (Gu et al., 2024) embeds a minimax criterion in the diffusion training process. It balances representativeness and diversity while reducing computational costs. IGD (Chen et al., 2025) redefines DD as a controlled diffusion generation task. It uses a trajectory influence function to guide the diffusion process and generate effective synthetic data for training.

### 2.2 Dataset Quantization

Traditional DD methods often rely on specific network architectures, resulting in degraded performance when the distilled datasets are transferred to other architectures (Wang et al., 2018; Zhao & Bilen, 2021b; Cazenavette et al., 2022). To address this limitation, Dataset Quantization (DQ) (Zhou et al., 2023) introduces a novel framework capable of compressing large-scale datasets and generating compact datasets suitable for training any neural network architecture without sacrificing performance. DQ achieves this by recursively partitioning the dataset into non-overlapping data bins and optimizing sample selection based on submodular gains to maximize dataset diversity. This approach not only enhances dataset diversity but also ensures the broad applicability of the generated subsets across varying network architectures. In a parallel effort to enhance both the diversity and realism of distilled data, Sun et al. (2024) introduces RDED, an efficient optimization-free paradigm that synthesizes new images by selecting and stitching patches from

real data. Zhao et al. (2024) further advanced this paradigm by proposing DQAS, an adaptive sampling strategy integrating active learning to refine class-imbalanced sample selection. This methodology addresses the limitations of traditional DQ approaches that assume uniform class distributions, enabling more efficient knowledge transfer across heterogeneous network architectures. Similarly, Li et al. (2025) proposes ADQ to improve naive DQ by adaptively sampling data based on the representativeness, diversity, and importance scores of the generated bins.

## 3 Method

### 3.1 Image Processing with Latent Diffusion

Diffusion models learn to generate images by iterative denoising a corrupted input through a sequence of timesteps that estimate the distribution of training data (Dhariwal & Nichol, 2021; Rombach et al., 2022; Saharia et al., 2022; Zhang et al., 2023). For image processing tasks, the latent diffusion model (Rombach et al., 2022) is conditioned on a provided text prompt $y$. The training objective minimizes the latent diffusion loss:

$$L = \mathbb{E}_{\mathcal{E}(x),y,\epsilon\sim\mathcal{N}(0,1),t}[\|\epsilon - \epsilon_\theta\left(z_t,t,\tau_\theta(y)\right)\|_2^2], \tag{1}$$

where $\mathcal{E}(x)$ and $\tau_\theta(y)$ represent the latent embeddings of the target image $x$ and conditional prompt $y$, encoded by a pretrained autoencoder $\mathcal{E}$ and CLIP $\tau_\theta$ (Radford et al., 2021). At each timestep, noise $\epsilon$ is added to the latent $z_t$, and the denoising network $\epsilon_\theta$ learns to predict the noise conditioned on $y$.

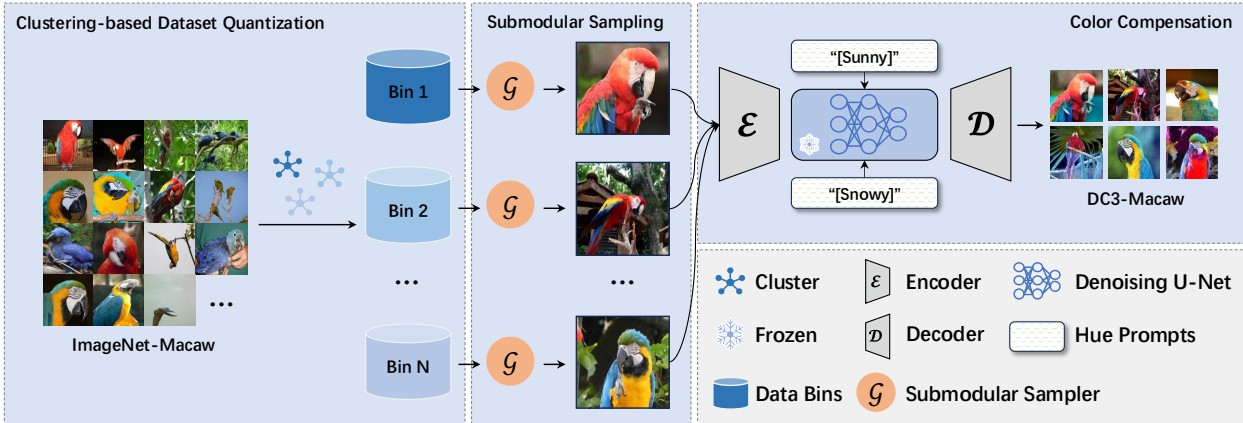

Figure 3: **Pipeline of the DC3 framework.** DC3 supports size-scalable, resolution-variable, and architecture-adaptive dataset condensation for downstream tasks and even fine-tuning large vision models (LVMs).

### 3.2 Color Compensation for Dataset Condensation

Our paper focuses on enhancing the color expression of condensed images. As illustrated in fig. 4, conventional image processing pipelines (Plataniotis & Venetsanopoulos, 2000; Jähne, 2005) suffer from color distortions due to insufficient semantic awareness and color reasoning capabilities. Most of the methods are implemented through mathematical transformations (Gonzalez, 2009). In contrast, diffusion-based image processing achieves pixel-level controllability through semantic-aware guidance, enabling a precise understanding and manipulation of color compensation, such as illumination adjustment and shadow harmonization. We provide a guideline for hue prompt selection in the supplementary materials.

Inspired by aesthetic adjustments in photographic post-processing, we propose **_Color Compensation_**, a mechanism integrating hue-related instructions into latent diffusion models to improve color diversity, as well as alleviate the **_Color Homogenization_** during dataset condensation. Specifically, what we have are conditional images $c_I$ and a set of hue prompts: $\mathcal{P} = \{c_1, c_2, \dots, c_L\}$. During compensation, the conditional

image $c_I$ is fed into the pretrained latent diffusion model $\Phi$ along with a randomly selected hue prompt $c_l$. The compensated image $I_c$ is defined as:

$$I_c = \Phi(c_I, c_l). \tag{2}$$

A hue is the dominant color family of a particular color. In DC3, the hue prompts are categorized into cool (e.g., rainy, snowy) and warm (e.g., sunny, daylight), except for the neutral hues (e.g., black, white, and gray). We select two hues from each group: $(\mathcal{P}_{\texttt{cool}}, \mathcal{P}_{\texttt{warm}})$, and implement the color compensation to synthesize $(I_{\texttt{cool}}, I_{\texttt{warm}})$ according to eq. (2). Please refer to section C for a detailed guideline on prompt selection. Furthermore, we crop them in half and then stitch them to create a more informative image.

As illustrated in fig. 4, ***color compensation*** effectively enhances dataset colorfulness by expanding the color gamut representation without introducing artificial artifacts, thereby contributing to model training on downstream tasks. Compared to RDED (Sun et al., 2024), another crop-and-stitch approach that composites images through disparate image regions, our method synthesizes multiple color-compensated variants of identical images. This strategy preserves structural consistency while promoting generalized representations across diverse visual domains, such as resolution and color distribution.

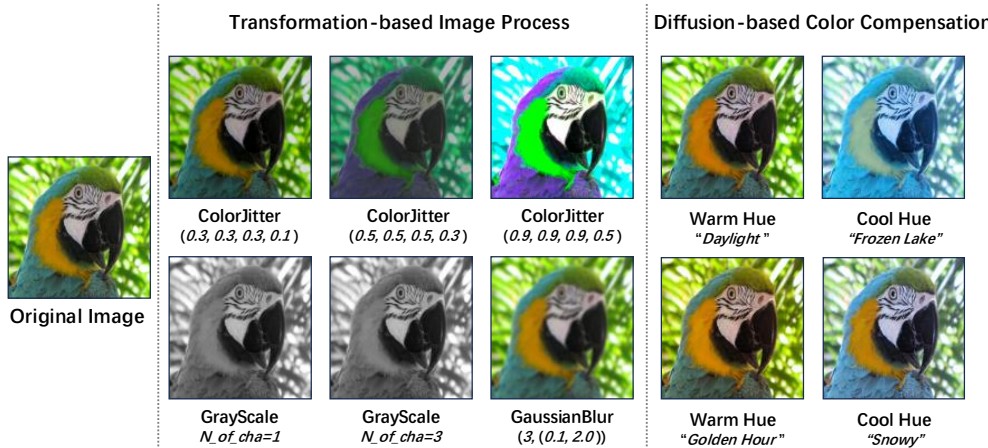

Figure 4: **Comparison between conventional and diffusion-based image processes.** Left: Original image. Middle: Visualizations of conventional processing methods: (a) ColorJitter (brightness, contrast, saturation, hue), (b) Grayscale conversions (number of output channels), and (c) Gaussian blur (kernel, $\sigma$). Right: Diffusion-based color compensation. The diffusion-based approach achieves semantic-aware color reasoning, mitigating distortions caused by mathematical transformations in traditional pipelines.

### 3.3 Quantization Analysis

DQ is an emerging method for dataset condensation based on image-level selection. The idea of DQ is to partition data into different bins iteratively based on submodular gain, which was first proposed in GraphCut (Iyer et al., 2021) for coreset selection. The samples are selected from the bins randomly, whose distribution is aligned with the original datasets, i.e., it does not have the ***color homogenization*** problem. Nevertheless, a fundamental limitation of this approach lies in its inherently monotonic decay of submodular gains across increasing bin indices, which inevitably causes random sampling to discard critical samples essential for model generalization. To reveal the relationship between submodular gain and sample distribution, we visualize the samples in ImageNette according to the values of their submodular gains with the help of t-SNE (Van der Maaten & Hinton, 2008) plots (the mid-row of fig. 5). Also, we provide additional visualizations for the first, second, and last samples selected by the submodular gain criterion (the top row and bottom row of fig. 5).

Due to DQ performing random sampling within a certain bin, the following situations exist: (a) Samples with high submodular gains may be excluded despite their distance from the optimal sample that has the

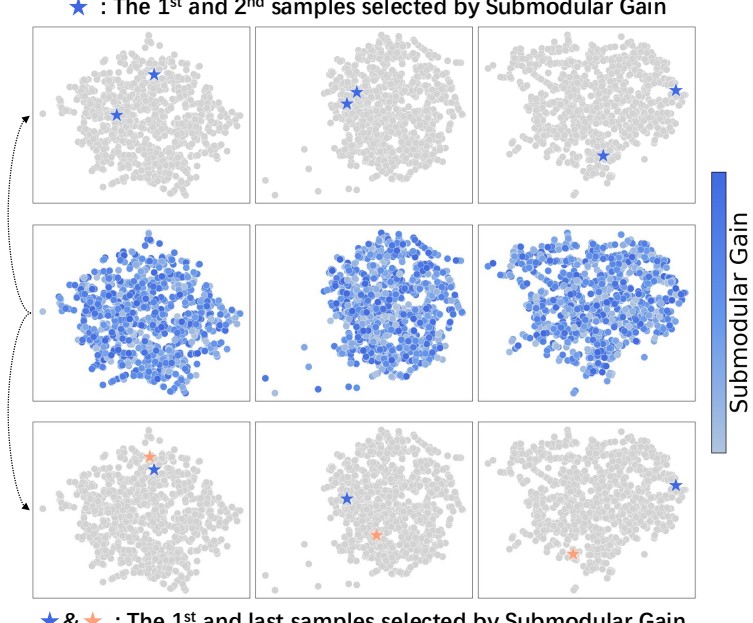

★ : The 1st and 2nd samples selected by Submodular Gain

★ & ★ : The 1st and last samples selected by Submodular Gain

Figure 5: **Visualizations of the relationship between submodular gain and sample distribution.** The following situations occur in traditional DQ methods: (a) inconsequential but compact samples may be selected because they are in different bins (bottom-left), and (b) essential and diverse samples may be discarded because they are in the same bins (top-right). Each column represents a different category.

highest submodular gain (top-right in fig. 5). (b) Samples with low submodular gains may be selected even when proximate to the optimal sample (bottom-left in fig. 5). Based on the visualization, we argue that the submodular gain of a sample is independent of its position in the feature distribution. In other words, the sampling diversity does not necessarily benefit submodular gain.

To avoid selecting samples with small gains, we propose clustering-based DQ, where the submodular gain is then used as an image sampler instead of a bin generator. Specifically, the clustering algorithm divides the dataset into multiple clusters, which are naturally regarded as data bins ($C_j$). Each sample $x$ is iteratively assigned to the closest bins:

$$\arg\min_{j} \|x - C_j\|^2, \ j = 1, \ldots, M. \tag{3}$$

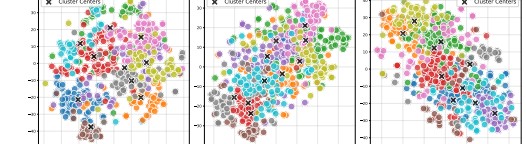

As shown in fig. 6, data bins, also known as clusters, are dispersed throughout the feature space as expected. The clustering algorithm performs a coarse partitioning of samples based on feature-space similarities. Within each bin, we leverage submodular sampling to identify the representative samples with high submodular gains. In this way, the condensed dataset preserves maximum semantic and feature diversity.

Figure 6: **T-SNE visualizations of intra-class clusters on ImageNet.** Submodular sampling selects the representative samples to preserve maximum feature diversity. Each column represents a different category.

### 3.4 Submodular Sampling

***Color compensation*** enhances the diversity of the images, but we need to know which images require compensation, as compensating the entire dataset is computational. Thus, we introduce submodular sampling as a feasible solution. The clustering-based DQ naturally separates the intra-class structure into different sample bins. Then, with the help of submodular sampling, we can select significant and diverse samples from the class.

The samples with high submodular gain ($G(x_k)$) will be sampled according to:

$$G(x_k) = \sum_{p \in S^{k-1}} \|f(p) - f(x_k)\|_2^2 - \sum_{p \in \bar{S}^{k-1}} \|f(p) - f(x_k)\|_2^2, \qquad (4)$$

where $f(p)$ is the feature of already selected sample $p$ and $f(x_k)$ is the feature of the current sample $x_k$. $S^{k-1}$ is the sample set selected from the whole dataset $D$ and $\bar{S}^{k-1} = D \backslash S^{k-1}$ is the complementary set. The submodular sampling is summarized in algorithm 1.

---

**Algorithm 1** Submodular Sampling

---

**Input:** $x \in C$: Data in bins (Bins generated by clustering).
**Input:** $M$: Number of data bins.
**Input:** $N$: Image per class (IPC).
1: **for** $C_j$ in $C$ **do**
2:     **for** $x_k$ in $C_j$ **do**
3:         Compute $G(x_k)$ according to eq. (4)
4:     **end for**
5:     $S_j = \emptyset$
6:     $C'_j = \text{sort}(C_j, G, \texttt{desc})$
7:     **if** $N \leq M$ **then**
8:         $M = N$
9:     **end if**
10:     $S_j = C'_j[0 : N/M]$                               # Selection
11: **end for**
12: $S = \bigcup_{j=1}^{M} S_j$
**Output:** $S$: The selected sample set.

---

To sum up, DC3 quantifies sample-wise contribution through submodular gain and improves sample performance with ***Color Compensation***, ensuring that the compressed dataset remains both representative and diverse across categories. By effectively preserving critical information, our approach enhances the overall quality and utility of the condensed dataset. The entire DC3 pipeline is summarized in fig. 3.

## 4 Experiments

### 4.1 Experimental Settings

We perform systematic evaluations across varying-scale benchmarks to assess generalization boundaries. For large-scale datasets, we include ImageNet-1K (Deng et al., 2009) ($224 \times 224$) and its subsets, such as Tiny-ImageNet ($64 \times 64$). Small-scale low-resolution ($32 \times 32$) analysis utilizes CIFAR-10/100 (Krizhevsky et al., 2009). To quantify task difficulty sensitivity, we benchmark ImageNette and ImageWoof, the subsets of ImageNet with 10 classes. Note that ImageWoof poses greater discrimination challenges due to more inter-class similarity. We use Stable Diffusion-V1.5 and DiT-XL/2-256 as our foundation models.

Following the prior works (Sun et al., 2024; Chen et al., 2025), we set IPC to 1, 10, and 50. For a fair comparison, we adhere to the official implementation of RDED (Sun et al., 2024) and employ a ResNet-18 network to generate soft labels for fine-grained supervision. Performance validation is conducted using PyTorch on $8\times$ NVIDIA 3090 GPUs. Detailed settings are discussed in the supplementary materials.

### 4.2 Comparison with SOTA Methods

While our method draws inspiration from DQ, we exclude direct comparisons due to differing evaluation priorities. Specifically, DQ emphasizes generalization across large IPC keep ratios, whereas our focus lies in optimizing compression performance through color compensation at low IPC regimes. Consequently, we benchmark against DD such as the matching-based TESLA (Cui et al., 2023), SRe$^2$L (Yin et al., 2023),

DataDAM (Sajedi et al., 2023), RDED (Sun et al., 2024), and generation-based Minimax (Gu et al., 2024), $D^4M$ (Su et al., 2024), and IGD (Chen et al., 2025), aligning with the objective of DC that maximizes dataset utility under extreme compression. Importantly, our DC3 inherits all the advantages of DQ while achieving superior compression efficiency.

**ImageNet family.** The extension of dataset condensation to large-scale scenarios represents a critical advancement in DCAI research. Our method demonstrates superior scalability and effectiveness on this challenging task. As evidenced by table 1, DC3 achieves SOTA performance across all IPC settings on large-scale ImageNet-1K. Notably, DC3 delivers an 8.4% Top-1 accuracy improvement over RDED at IPC = 10, surpassing the SOTA results on a large-scale dataset. This performance gap underscores its superior ability to represent complex visual patterns, particularly in high-resolution large-category scenarios.

| Dataset | IPC | Matching-based | | | | Generation-based | | DC3 | Full |
| | | TESLA[†] | SRe$^2$L | RDED[†] | Minimax | $D^4M$ | IGD | | |
|---|---|---|---|---|---|---|---|---|---|
| ImageNet-1K | 1 | $7.7_{\pm 0.2}$ | $0.1_{\pm 0.1}$ | $6.6_{\pm 0.2}$ | - | - | - | $\mathbf{8.1_{\pm 0.3}}$ | |
| | 10 | $17.8_{\pm 1.3}$ | $21.3_{\pm 0.6}$ | $42.0_{\pm 0.1}$ | $44.3_{\pm 0.5}$ | $27.9_{\pm 0.9}$ | $46.2_{\pm 0.6}$ | $\mathbf{50.4_{\pm 0.2}}$ | 69.8 |
| | 50 | $27.9_{\pm 1.2}$ | $46.8_{\pm 0.2}$ | $56.5_{\pm 0.1}$ | $58.6_{\pm 0.3}$ | $55.2_{\pm 0.4}$ | $60.3_{\pm 0.4}$ | $\mathbf{62.3_{\pm 0.7}}$ | |

Table 1: **Top-1 Accuracy↑ on ImageNet-1K.** The results of DC3 and other state-of-the-art methods are evaluated on ResNet-18. †: The results of TESLA are evaluated on ConvNet.

Furthermore, we validate DC3 efficacy on some challenging domain-specific ImageNet subsets. As listed in table 2, our approach achieves a 16.4% accuracy gain over IGD at IPC-10 on ImageNette. For the more complex Tiny-ImageNet (200 categories), our method has better performance than others, demonstrating consistent performance across varying task complexities.

| Dataset | IPC | RDED | Minimax | $D^4M$ | IGD | DC3 |
|---|---|---|---|---|---|---|
| ImageWoof | 1 | $20.8_{\pm 1.2}$ | - | - | - | $\mathbf{23.2_{\pm 1.4}}$ |
| | 10 | $38.5_{\pm 2.1}$ | $37.6_{\pm 0.9}$ | $42.9_{\pm 1.1}$ | $47.2_{\pm 1.6}$ | $\mathbf{48.7_{\pm 0.1}}$ |
| | 50 | $68.5_{\pm 0.7}$ | $57.1_{\pm 0.6}$ | $62.1_{\pm 0.4}$ | $65.4_{\pm 1.8}$ | $\mathbf{72.4_{\pm 0.6}}$ |
| ImageNette | 1 | $35.8_{\pm 1.0}$ | $32.1_{\pm 0.3}$ | $34.1_{\pm 0.9}$ | - | $\mathbf{37.6_{\pm 0.9}}$ |
| | 10 | $61.4_{\pm 0.4}$ | $62.0_{\pm 0.2}$ | $68.4_{\pm 0.8}$ | $66.2_{\pm 1.2}$ | $\mathbf{84.8_{\pm 1.1}}$ |
| | 50 | $80.4_{\pm 0.4}$ | $76.6_{\pm 0.2}$ | $81.3_{\pm 0.1}$ | $82.0_{\pm 0.3}$ | $\mathbf{89.8_{\pm 0.2}}$ |
| Tiny-ImageNet | 1 | $9.7_{\pm 0.4}$ | $13.3_{\pm 0.8}$ | $15.1_{\pm 0.2}$ | - | $\mathbf{20.0_{\pm 1.2}}$ |
| | 10 | $41.9_{\pm 0.2}$ | $39.2_{\pm 0.7}$ | $35.7_{\pm 0.6}$ | - | $\mathbf{45.1_{\pm 1.1}}$ |
| | 50 | $58.2_{\pm 0.1}$ | $44.8_{\pm 0.2}$ | $46.2_{\pm 1.1}$ | - | $\mathbf{59.4_{\pm 0.7}}$ |

Table 2: **Top-1 Accuracy↑ on ImageNet subsets: ImageWoof, ImageNette, and Tiny-ImageNet.** All of the results are evaluated by ResNet-18. DC3 holds the optimal performance among datasets and methods.

The ImageWoof subset, known for its high intra-class similarity, poses challenges for dataset condensation. Nonetheless, our method achieves the best performance on this benchmark. The success can be attributed to two principal factors: (1) our ***color compensation*** mechanism effectively preserves subtle discriminative features that are critical for distinguishing similar classes, and (2) the submodular sampling strategy ensures optimal coverage of diverse intra-class variations, preventing the loss of essential patterns during condensation.

**CIFAR-10/100.** To assess the generalization across varied scales and resolutions, we also evaluate DC3 on CIFAR-10 and CIFAR-100. As shown in table 3, our approach exceeds those of previous SOTA models by a significant margin on the small-scale benchmarks. The accuracy at IPC-10 is 1.5× of our baseline

RDED. For the more challenging CIFAR-100, a 100-class fine-grained dataset, our method maintains strong competitiveness across different IPCs, demonstrating robustness to both coarse and fine-grained classification tasks at low resolutions.

| Dataset | IPC | RDED | SRe$^2$L | DataDAM | D$^4$M | DC3 |
|---------|-----|------|----------|---------|--------|-----|
| CIFAR-10 | 1 | $22.9_{\pm0.4}$ | $16.9_{\pm0.9}$ | $\mathbf{32.0}_{\pm1.2}$ | $17.6_{\pm1.1}$ | $25.6_{\pm1.2}$ |
| | 10 | $37.1_{\pm0.3}$ | $27.2_{\pm0.5}$ | $\underline{54.2}_{\pm0.8}$ | $51.5_{\pm0.5}$ | $\mathbf{57.8}_{\pm0.8}$ |
| | 50 | $62.1_{\pm0.1}$ | $47.5_{\pm0.6}$ | $\underline{67.0}_{\pm0.4}$ | $62.3_{\pm0.2}$ | $\mathbf{80.9}_{\pm0.5}$ |
| CIFAR-100 | 1 | $11.0_{\pm0.3}$ | $2.0_{\pm0.2}$ | $\underline{14.5}_{\pm0.5}$ | $5.7_{\pm1.4}$ | $\mathbf{21.1}_{\pm1.6}$ |
| | 10 | $42.6_{\pm0.2}$ | $23.5_{\pm0.8}$ | $\underline{34.8}_{\pm0.5}$ | $44.0_{\pm0.3}$ | $\mathbf{57.4}_{\pm0.2}$ |
| | 50 | $\underline{62.6}_{\pm0.1}$ | $51.4_{\pm0.8}$ | $49.4_{\pm0.3}$ | $48.4_{\pm0.9}$ | $\mathbf{64.2}_{\pm0.5}$ |

Table 3: **Top-1 Accuracy↑ on CIFAR-10 and CIFAR-100.** The results are evaluated by ResNet-18.

Last but not least, we find that the existing dataset condensation methods exhibit varying degrees of performance degradation under different *scales*, *class numbers*, and *structures*. DC3 enhances the data colorfulness and inherits the generalization performance of DQ, successfully overcoming this adaptation challenge.

### 4.3 Cross-architecture Generalization

To support the claim that pixel optimization methods, which cause "color homogenization", impair model generalization, a direct comparison is made between DC3 and various matching-based DD methods (such as KIP (Nguyen et al., 2021b), DSA) on "Seen" and "Unseen" network architectures. The results in table 4 indicate that the aforementioned methods experience a sharp accuracy decrease when evaluated on architectures not previously seen. Such significant performance degradation provides strong empirical support for the hypothesis that "color homogenization" hinders model learning of transferable data representations. Among all methods, DC3 and DSA demonstrate the most robust generalization to the unseen architecture, exhibiting the slightest performance degradation. Crucially, DC3 absolute performance has far exceeded that of other methods.

| Eval. Model | KIP | DSA | MTT | TESLA | DC3 |
|-------------|-----|-----|-----|-------|-----|
| Seen | 47.2 | 53.0 | 65.3 | 66.4 | $\mathbf{84.8}_{\pm1.2}$ |
| Unseen | $15.9_{\downarrow31.3}$ | $31.9_{\downarrow21.1}$ | $34.6_{\downarrow30.7}$ | $34.8_{\downarrow31.6}$ | $\mathbf{60.4}_{\pm1.2\downarrow24.4}$ |

Table 4: **Different DD methods show performance degradation on the "unseen" architecture.** Here, "seen" architectures refer to CNN-based models (e.g., ResNet-18 or ConvNet-D6), while "unseen" architectures are Transformer-based (e.g., Swin-V2-T or ViT-S). We choose the worst result of DC3 on Swin-V2-T under IPC-10, but it still performs the best among methods.

To evaluate the cross-architecture generalization of compressed datasets, we deploy them to CNN-based (ResNet series, MobileNet-V2, EfficientNet-B0) and Transformer-based architecture (Swin-V2-T). We employ ResNet-18 as the feature sampler during submodular sampling, thus, we make it the teacher model naturally. As demonstrated in table 5, ResNet-18 achieves 84.8% Top-1 accuracy at IPC-10, while Swin-V2-T exhibits a significant drop to 60.4%. This discrepancy stems from fundamental architectural divergences: CNNs rely on hierarchical feature extraction through localized receptive fields, whereas Transformers model the long-range dependencies via global attention mechanisms. Such structural differences lead to varying sensitivities to high-frequency details in compressed data. Despite inherent architectural biases, Swin-V2-T achieves 80.0% accuracy at IPC-50, marking a 19.6% improvement over IPC-10, which highlights its potential for multi-architecture applications.

Notably, the performance volatility (standard deviation) is approximately 3.93% (IPC-50) across models ranging from lightweight MobileNet-V2 to deep ResNet-101, demonstrating its robustness to model scale.

| Sampler | IPC | ResNet-18 | ResNet-50 | ResNet-101 | MobileNet-V2 | EfficientNet-B0 | Swin-V2-T |
|---------|-----|-----------|-----------|------------|--------------|-----------------|-----------|
| ResNet-18 | 10 | $\mathbf{84.8}_{\pm 1.2}$ | $80.8_{\pm 1.5}$ | $79.0_{\pm 1.3}$ | $81.8_{\pm 1.1}$ | $\underline{84.2}_{\pm 1.4}$ | $60.4_{\pm 1.2}$ |
|  | 50 | $\underline{89.8}_{\pm 0.3}$ | $89.0_{\pm 0.2}$ | $89.2_{\pm 0.4}$ | $89.6_{\pm 0.5}$ | $\mathbf{90.2}_{\pm 0.4}$ | $80.0_{\pm 0.5}$ |

Table 5: **Generalization performance of DC3 across different neural network architectures.** The results demonstrate the robustness and adaptability of the condensed datasets to various models.

Additionally, DC3 achieves the best accuracy of 90.2% on EfficientNet-B0, which also validates its alignment with efficient network designs. These findings not only confirm the cross-architecture and cross-scale generalization of DC3 but also elucidate the coupling mechanism between dataset compression and model inductive bias.

### 4.4 Ablation Study

**Color compensation.** This section addresses a pivotal question: *Does the powerful distillation capability of DC3 stem from the inherent data augmentation ability of the diffusion model itself, or from the specific **color compensation** strategy designed to address the **color homogenization** problem?* For this purpose, we design a series of ablations on diffusion model schemes with three different kinds of prompts. The first is a "null" prompt, providing no

| Dataset | IPC | Null | HR | DC3 |
|---------|-----|------|-----|-----|
| CIFAR-10 | 10 | $50.3_{\pm 0.2}$ | $54.5_{\pm 0.2}$ | $\mathbf{57.8}_{\pm 0.8}$ |
|  | 50 | $73.6_{\pm 0.1}$ | $76.0_{\pm 0.2}$ | $\mathbf{80.9}_{\pm 0.5}$ |
| CIFAR-100 | 10 | $51.8_{\pm 0.1}$ | $53.5_{\pm 0.2}$ | $\mathbf{57.4}_{\pm 0.2}$ |
|  | 50 | $58.3_{\pm 0.1}$ | $61.2_{\pm 0.1}$ | $\mathbf{64.2}_{\pm 0.5}$ |

Table 6: Comparison results of different prompts.

information as a performance baseline. The second is the "HR" prompt, which uses "**H**igh-**R**esolution images" as prompt words to evaluate the inherent effectiveness of the diffusion model itself. The third is the carefully designed DC3 color compensation prompt. A guideline on how to select DC3 prompts is provided in section C. The results in table 6 demonstrate that the DC3 strategy achieves optimal performance on both the CIFAR-10 and CIFAR-100 datasets, regardless of the IPC settings, which confirms that **the performance improvement is not simply due to the introduction of diffusion models, but attributed to the proposed color compensation strategy.** This strategy effectively enhances the color diversity of the compressed data, solving the common issue of ***color homogenization*** in previous methods and improving the final performance of the surrogate model.

**Compensation manners.** To further validate the effectiveness of ***color compensation***, we introduce the *colorfulness* metric (Hasler & Suesstrunk, 2003), which reflects the color richness in an image and helps establish a correlation between experimental metrics and the color attributes.

Specifically, through the red-green (rg) and yellow-blue (yb) difference of an image, the *colorfulness* can be defined as:

$$colorfulness = \sigma_{\text{root}} + 0.3 \times \mu_{\text{root}}, \quad (5)$$

where $\sigma_{\text{root}}$ and $\mu_{\text{root}}$ represent the combined standard deviation and mean of the rg and yb differences, respectively:

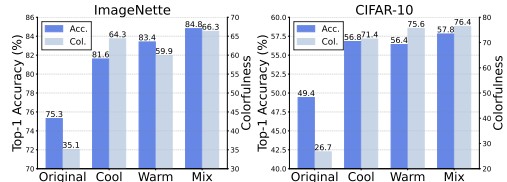

Figure 7: **Ablation study on color compensation.** The colorfulness and accuracy results highlight the enhancement in color diversity with the proposed diffusion-based color compensation technique.

$$\sigma_{\text{root}} = \sqrt{\sigma_{\text{rg}}^2 + \sigma_{\text{yb}}^2}, \quad \mu_{\text{root}} = \sqrt{\mu_{\text{rg}}^2 + \mu_{\text{yb}}^2}. \quad (6)$$

In eq. (6), rg $= |R - G|$ and yb $= 0.5 \times (R + G) - B$ are the differences between the Red, Green, and Blue components of an image.

As depicted in fig. 7, the application of ***color compensation*** to submodular-sampled datasets demonstrates significant theoretical advantages over uncompensated approaches. The compensation strategy addresses the inherent color distribution imbalances in compressed datasets by enhancing the color of underrepresented regions while maintaining visual naturalness. The cool hue specifically targets cyan-blue-magenta deficiencies,

while warm compensation addresses yellow-orange-red limitations. Thus, the mixed compensation provides a comprehensive spectral coverage. Section D discusses the different mix methods in detail.

**Compensation *v.s.* augmentation.** To comprehensively evaluate the dual contributions of our proposed DC3 framework in ***color compensation*** and coreset selection, a detailed ablation study is conducted with results listed in table 7. Since one of the primary contributions of DC3 is addressing the issue of ***color homogenization*** in existing methods, table 7 directly compares DC3 against other color augmentation techniques such as Color Jitter, HSV Shift, Balance Tuning, and AutoPalette (Yuan et al., 2024). The results demonstrate that DC3 significantly outperforms these alternative color solutions on the CIFAR-10 and CIFAR-100 datasets, thus confirming the unique advantage of the diffusion-based approach in enhancing color diversity. We named this strategy ***compensation*** instead of augmentation to distinguish it.

| Dataset | Color Jitter | HSV Shift | Balance Tuning | AutoPalette | DC3 |
|---|---|---|---|---|---|
| CIFAR-10 | $73.5_{\pm 1.8}$ | $74.2_{\pm 1.1}$ | $74.6_{\pm 1.4}$ | $79.4_{\pm 2.2}$ | $\mathbf{80.9}_{\pm 0.5}$ |
| CIFAR-100 | $53.2_{\pm 1.6}$ | $53.7_{\pm 1.4}$ | $54.4_{\pm 1.7}$ | $53.3_{\pm 1.3}$ | $\mathbf{64.2}_{\pm 0.8}$ |

Table 7: Comparison with augmentation methods under IPC-50. For fair comparison, we substitute Colour Compensation with augmentation methods, while the rest of the pipeline remains identical.

**Submodular sampling.** The upper section of table 8 reveals that DC3 with clustering-based submodular sampling performs better across CIFAR-10 and ImageNette under varying IPCs. The lower section only changes the sampling strategy, while the rest of the components remain fixed, including color compensation.

The competitive results of DC3 are attributed to the synergistic combination of clustering and submodular sampling. Clustering-based bin generation creates coherent and representative data bins by minimizing intra-cluster distances, providing a stronger foundation for the subsequent sampling process. Submodular sampling then optimally selects diverse and representative samples from these well-formed bins, capturing the underlying data distribution while maintaining diversity. This integration addresses the challenge in dataset condensation: preserving essential information while significantly reducing dataset scale.

| Method | Color Compensation | CIFAR-10 | ImageNette | CIFAR-10 | ImageNette |
|---|---|---|---|---|---|
| | | IPC = 10 | | IPC = 50 | |
| DQ | ✗ | $35.7_{\pm 0.7}$ | $69.3_{\pm 0.5}$ | $50.5_{\pm 0.4}$ | $74.2_{\pm 0.5}$ |
| DC3 | | $\mathbf{49.4}_{\pm 0.7}^{\uparrow 13.7}$ | $\mathbf{75.3}_{\pm 0.6}^{\uparrow 6.0}$ | $\mathbf{71.4}_{\pm 0.5}^{\uparrow 20.9}$ | $\mathbf{79.6}_{\pm 0.4}^{\uparrow 5.4}$ |
| DQ | ✔ | $51.2_{\pm 0.6}$ | $81.8_{\pm 1.5}$ | $68.4_{\pm 0.8}$ | $85.3_{\pm 0.6}$ |
| DC3 | | $\mathbf{57.8}_{\pm 0.8}^{\uparrow 6.6}$ | $\mathbf{84.8}_{\pm 1.1}^{\uparrow 3.0}$ | $\mathbf{80.9}_{\pm 0.5}^{\uparrow 12.5}$ | $\mathbf{89.8}_{\pm 0.2}^{\uparrow 4.5}$ |

Table 8: **Ablation study on sampling strategies.** DQ randomly selects samples within data bins, while DC3 utilizes submodular gain as the sampler.

Meanwhile, to validate the effectiveness of the improved submodular sampling selection strategy, table 9 compares DC3 with various commonly used coreset methods, including Herding, k-CG, GradMatch, and submodular-based DQ methods. The results show that DC3 achieved the best Top-1 accuracy on all tested datasets, particularly outstanding on CIFAR-10 (80.9%), CIFAR-100 (64.2%), and Tiny ImageNet (59.4%).

Overall, the above ablation experiments demonstrate that both color compensation and clustering-based submodular sampling in the DC3 framework are superior to existing technologies. It not only effectively improves the information density and representation ability of the compressed dataset but also provides an effective solution to the key bottleneck in dataset condensation.

**Pipeline analysis.** The ablation study on the DC3 pipeline reveals significant performance improvements by combining color compensation and submodular sampling. As listed in table 10, standalone submodular sampling improves accuracy by 10.9%, while color compensation achieves 15.1% on CIFAR-10. Their synergistic integration yields a 19.3% improvement. Similar trends hold on ImageNette.

| Method | CIFAR-10 | CIFAR-100 | Tiny-ImageNet |
|---|---|---|---|
| Herding | 34.8 | 34.4 | 45.7 |
| k-CG | 31.1 | 42.8 | 46.3 |
| GradMatch | 30.8 | 42.8 | 43.2 |
| DQ (Submodular) | $50.5_{\pm0.8}$ | $45.9_{\pm0.7}$ | $52.8_{\pm1.3}$ |
| DC3 | $\mathbf{80.9}_{\pm0.5}$ | $\mathbf{64.2}_{\pm0.5}$ | $\mathbf{59.4}_{\pm0.7}$ |

Table 9: Comparison with other selection methods under IPC-50.

This synergy arises from complementary mechanisms: color compensation mitigates chromatic homogenization by restoring natural hue distributions, while submodular sampling maximizes feature diversity. The former ensures perceptual fidelity critical for color-sensitive tasks, whereas the latter optimizes representativeness by prioritizing samples that span the data manifold. Their joint operation enforces a compressed dataset that aligns with both low-level chromatic statistics and high-level semantic structures, thereby bridging the gap between pixel-level optimization and image-level selection.

| S | C | CIFAR-10 | ImageNette |
|---|---|---|---|
| - | - | $38.5_{\pm0.1}$ | $55.8_{\pm0.1}$ |
| ✓ | - | $49.4_{\pm0.3}{}^{\uparrow10.9}$ | $75.3_{\pm0.4}{}^{\uparrow19.5}$ |
| - | ✓ | $53.6_{\pm0.6}{}^{\uparrow15.1}$ | $77.9_{\pm0.9}{}^{\uparrow22.1}$ |
| ✓ | ✓ | $\mathbf{57.8}_{\pm0.8}{}^{\uparrow19.3}$ | $\mathbf{84.8}_{\pm1.1}{}^{\uparrow29.0}$ |

Table 10: Ablation study on the impact of submodular sampling (S) and color compensation (C) strategies.

### 4.5 Training diffusion models with DC3

Besides being effective on downstream tasks, we argue that the condensed datasets should adapt to generation tasks. We fine-tune the Stable Diffusion (SD) and Diffusion Transformers (DiT) (Peebles & Xie, 2023) on the original and mixed (original and DC3-condensed IPC-50 data) ImageNette dataset with LoRA (Hu et al., 2022) and DiffFit (Xie et al., 2023).

The results drawn in fig. 8 show that datasets generated by DC3 effectively support the fine-tune pipelines without model collapse. Specifically, the models fine-tuned by mixed data achieve better FID-IS (Salimans et al., 2016; Heusel et al., 2017) scores compared to those on the original dataset and baselines, indicating better alignment with real data distributions. This

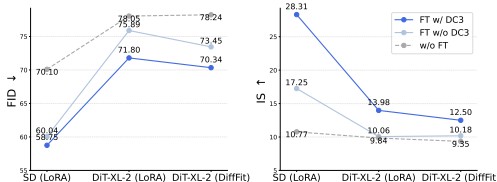

Figure 8: **Fine-tuning diffusion models with DC3.** To validate the effectiveness of our method, each fine-tuned model generates 10,000 images according to the number of samples in ImageNette (12894).

performance gap stems from its ability to preserve feature diversity by chromatic alignment and submodular sampling, which maintains essential data manifolds while removing redundant information.

## 5 Conclusion

We present DC3, a simple but efficient way to optimize both the performance and generalization of condensed datasets. Based on our color compensation mechanism, the experiments demonstrate favorable results across a wide range of datasets with various recognition difficulties. Notably, DC3 proves an information preservation mechanism within condensation, validating the feasibility of pre-training large vision models on condensed datasets without model collapse.

**Limitation and future work.** Although DC3 is a training-free approach, the compensation process is still constrained by the inference speed of the diffusion model. Future directions will focus on extremely efficient DC under extreme compression. We hope this work will inspire further research into dataset condensation. Perhaps in the future, the scaling law will be broken to a linear level with the help of Data-Centric AI.

## Acknowledgments

This research was supported in part by JSPS KAKENHI Grant Numbers JP24K23849 and JP25K21218.

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

## Appendix

## A    Experimental Details

The collaboration between sampling and compensation enables DC3 to possess powerful condensation capabilities. There are details of these pipelines that require explanation. In submodular sampling, section B presents the results for different cluster numbers (K), as K significantly affects the sampling results. Also, in the color compensation phase, section C introduces the selection guideline of hue prompts. Additionally, section D experiments on different crop-and-stitch methods to further enhance the information density for the condensed images.

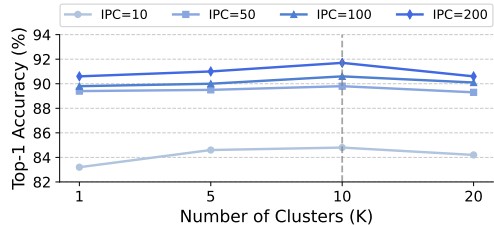

Figure 9: Top-1 Accuracy↑ of the number of clusters (K) for varying IPCs (IPC-10, 50, 100, 200). The results demonstrate the impact of cluster granularity on classification performance.

During the validation stage, the parameter details of all experiments are listed in table 11. Finally, we provide a wealth of visualizations in sections E, G and H, including condensed images and fine-tuned DiT-generated images, which prove the effectiveness of DC3. For DC3 and the reproduced baselines, we report the mean and standard deviation over three runs. For other competing methods, we quote the results directly from their original publications.

| Settings | Values |
|---|---|
| guidance scale | 4 |
| network | ResNet18 |
| input size | 224 |
| optimizer | AdamW |
| learning rate | 0.001 |
| weight decay | 0.01 |

(a) ImageNet

| Settings | Values |
|---|---|
| guidance scale | 4 |
| network | ResNet18 |
| input size | 32 |
| optimizer | AdamW |
| learning rate | 0.001 |
| weight decay | 0.01 |

(b) CIFAR-10 and CIFAR-100

| Settings | Values |
|---|---|
| guidance scale | 4 |
| network | ResNet18 |
| input size | 224 |
| optimizer | AdamW |
| learning rate | 0.001 |
| weight decay | 0.01 |

(c) ImageWoof and ImageNette

Table 11: Evaluation details for different datasets.

## B   Selection of Cluster Numbers

The number of clusters (K) should be chosen carefully. Thus, we conduct experiments using four clustering settings: K = 1, 5, 10, and 20. As depicted in fig. 9, the Top-1 accuracy reaches an optimal value at K = 10. According to fig. 9, we can also claim that DC3 is not sensitive to K, since the accuracy drops slightly at other K settings.

## C   Guideline of Hue Prompt Selection

DC3 attempted to implement pixel-level optimization to mitigate **color homogenization** using instruction-conditioned diffusion models. Cool and warm hues group the instructions. This taxonomy originates from quantitative analysis of natural lighting in photographic aesthetics: cool-hue scenes exhibit spectral energy concentrated in short wavelengths (400-500 nm), whereas warm hues correlate with long-wavelength distributions (550-700 nm). As listed in table 12, cool-hue prompts (e.g., *"rainy"*, *"snowy"*) align with low-color-temperature scenes (e.g., blue-cyan hue in rain/snow). In contrast, warm-hue prompts (e.g., *"sunny"*, *"golden hour"*) cap-

| Cool Hue | Warm Hue |
|---|---|
| *rainy* | *sepia* |
| *snowy* | *sunny* |
| *infrared* | *daylight* |
| *underwater* | *vivid colors* |
| *frozen lake* | *golden hour* |

Table 12: Examples of instruction prompts for cool and warm hues.

ture high-color-temperature spectral characteristics (e.g., orange-yellow hue under strong illumination). By embedding these physically grounded semantic instructions into latent diffusion models, we successfully implement pixel-level color compensation (e.g., adjusting illumination while preserving structural integrity).

The selection strategy adheres to the following criteria: (a) *visual discriminability* — terms must induce color shifts aligned with human perception (e.g., *"underwater"* implies blue-green hue, *"vivid colors"* triggers saturation enhancement), (b) *complementary coverage* — prompts should span opposing color families to maximize gamut diversity. In our experiment, we randomly selected one prompt from both cool and warm hues for color compensation. Then we crop-and-stitch the generated image to create a single distilled image.

## D   Ablation on Crop-and-Stitch

We appraise three kinds of crop-and-stitch methods on datasets with different resolutions. The experimental results listed in table 13 demonstrate that crop-and-stitch in half yields the best results across resolutions.

Excessive stitching (e.g., grid stitch and extreme pixel stitch) will degrade the semantic integrity, impairing feature extraction capabilities. Pixel stitch exhibits unstable performance due to insufficient spatial continuity constraints. Half merge effectively preserves structural semantics while enabling information fusion, particularly advantageous for cool/warm hue compensation tasks. By maintaining complete half-region structures, this method retains coherent color distribution patterns, thereby enhancing discriminative learn-

ing of chromatic features. The accuracy performance validates half-merge as the optimal strategy for image fusion in DC3.

| Dataset | IPC | $2 \times \frac{1}{2}$ | $4 \times \frac{1}{4}$ | 50% Pixels | $8 \times 8$ Grids | $16 \times 16$ Grids |
|---|---|---|---|---|---|---|
| CIFAR-10 | 1 | **25.6** | 22.4 | 24.9 | 23.0 | 19.5 |
| | 10 | 57.8 | **58.2** | 54.6 | 56.1 | 44.2 |
| ImageWoof | 1 | **15.2** | 12.6 | 12.8 | 15.0 | 10.2 |
| | 10 | **38.0** | 35.6 | 37.6 | 34.4 | 37.4 |
| ImageNette | 1 | **37.6** | 32.2 | 28.8 | 36.4 | 27.8 |
| | 10 | **84.8** | 81.7 | 76.4 | 75.8 | 77.1 |

Table 13: **Ablation results on different stitch methods.** $n \times \frac{1}{n}$: Crop $\frac{1}{n}$ of each of the $n$ compensated images and then stitch. Pixels: Randomly extract 50% of the pixels from 2 compensated images and then stitch. $n \times n$ Grids: Divide each of the 2 compensated images into $n \times n$ grids, and then randomly stitch half of the grids into a complete image.

# E    Visualization Analysis

As a performance matching method, the KDE curves of MTT are different from those of other matching methods. From fig. 10a, the distribution reveals a leptokurtic profile characterized by a peaked central tendency and attenuated tails. This statistical property indicates significant **Color Redundancy**, where the compressed dataset over-represents narrow chromatic ranges while under-sampling critical color variations. Both **Color Redundancy** and **Color Homogenization** phenomena compressed data manifolds inadequately span the original color space, thereby violating the manifold hypothesis essential for dataset generalization. In contrast, DC3 fits the color distribution well on CIFAR-10 (fig. 10b) and ImageNette (fig. 10c) and achieves superior performance.

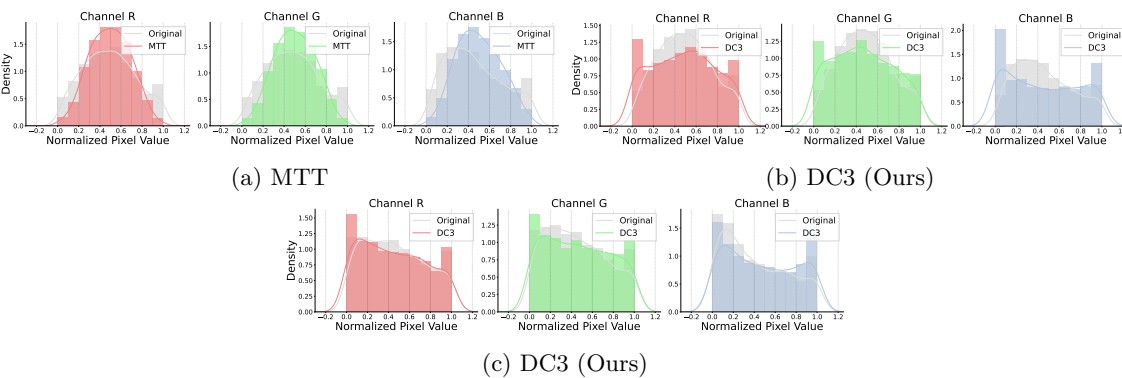

(a) MTT                                                                                    (b) DC3 (Ours)

(c) DC3 (Ours)

Figure 10: **The KDE curves of the normalized RGB pixel value of MTT and DC3.** (a)-(b) CIFAR-10 results on MTT and DC3. (c) ImageNette results on DC3.

# F    Comparison of Computation Time and GPU Memory Cost

Table 14 evaluates the computational efficiency of the DC3 framework, comparing its Color Compensation stage (left table) and Bin Generation stage (right table) against relevant state-of-the-art methods. The results indicate that even when processing high-resolution ($512 \times 512$) images, DC3's time consumption (4.1s) and GPU memory footprint (2.7GB) are significantly lower than those of methods such as TESLA and SRe$^2$L. Concurrently, the right table shows that its Bin Generation efficiency is marginally superior to the conventional DQ method. In summary, DC3 not only surpasses existing methods in performance but

also demonstrates a substantial advantage in computational cost, proving the framework's high practical feasibility and efficiency in achieving high-quality dataset compression.

| Method | Resolution | Time(s)↓ | GPU(GB)↓ | Method | GenBin(h)↓ |
|--------|-----------|----------|----------|--------|-----------|
| TESLA | $64 \times 64$ | 46.0 | 13.9 | DQ | $\sim 1.0$ |
| MTT | $128 \times 128$ | 45.0 | 79.9 | DC3 | $\sim \mathbf{0.9}$ |
| SRe$^2$L | $224 \times 224$ | 5.2 | 34.8 | | |
| DC3 | $512 \times 512$ | **4.1** | **2.7** | | |

Table 14: Comparison of computation time and GPU memory cost.

## G   Visualizations from Fine-tuned DiT

From the experimental results shown in the main text and fig. 11, we believe that DC3 does not cause model collapse after fine-tuning DiT for 50 epochs. Furthermore, using DC3-compressed data as an auxiliary training set yields semantic-consistent generation results.

This effectiveness stems from the ability to maintain fidelity and enhance the feature discriminability of DC3, which aligns the latent space of condensed data with natural image priors using diffusion models. This work also pioneers the direction of fine-tuning the generative models using compressed datasets. The across-architecture generalization results in the main text further validating the practicality of our approach for data-efficient transfer learning.

## H   More Visualizations

We randomly sample the images from the DC3 condensed CIFAR-10/100, ImageWoof, and ImageNette. The visualizations are depicted in figs. 12 to 15.

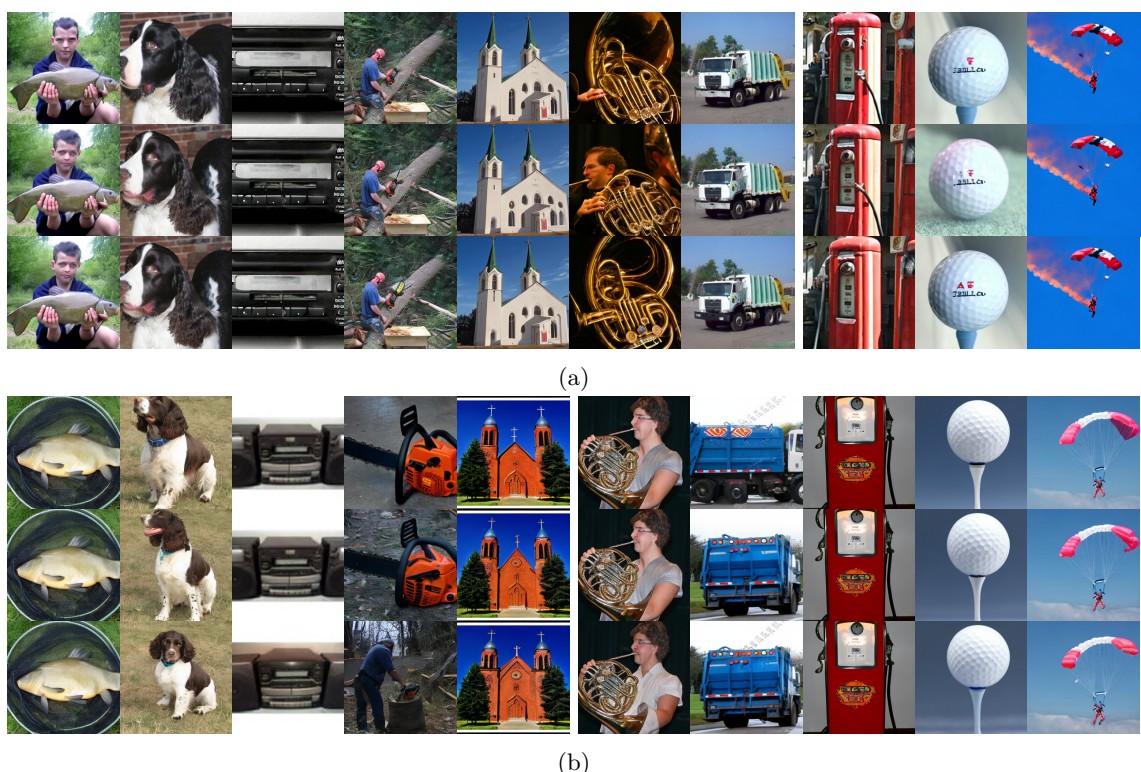

(a)

(b)

Figure 11: Comparison of images generated by ImageNette fine-tuned DiT with different settings. The first row displays outputs from DiT without fine-tuning. The second row visualizes the results from the model fine-tuned with the original data, while the third row presents outputs from the mixed data (original ImageNette and DC3 condensed ImageNette IPC-50) fine-tuned DiT.

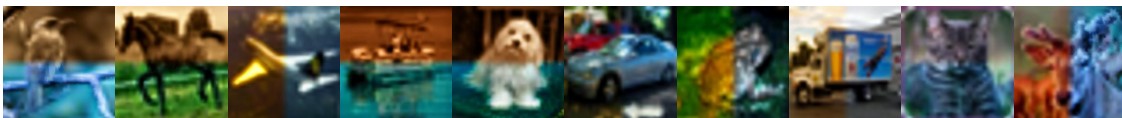

Figure 12: More visualizations selected from the condensed CIFAR-10.

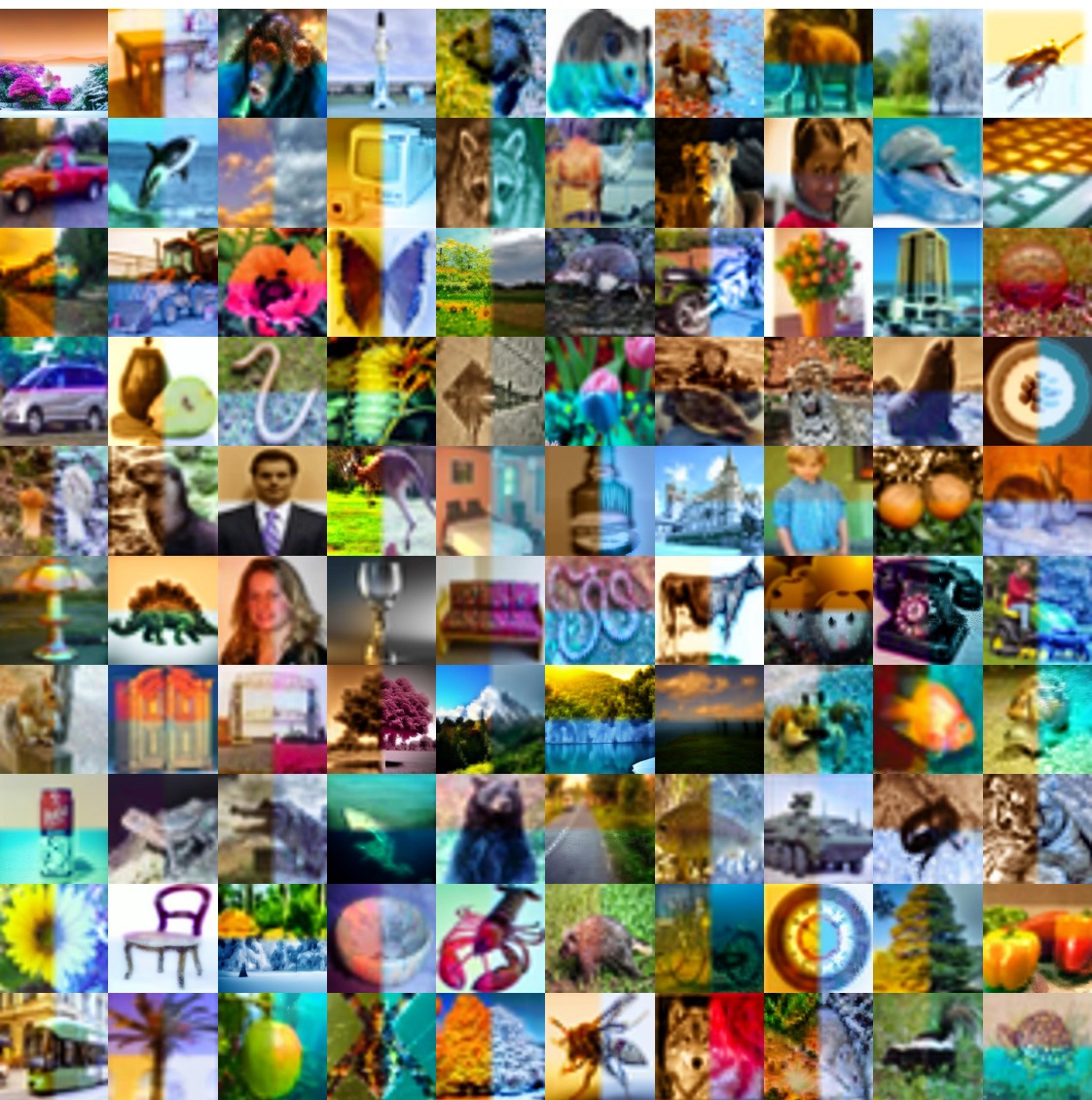

Figure 13: More visualizations selected from the condensed CIFAR-100.

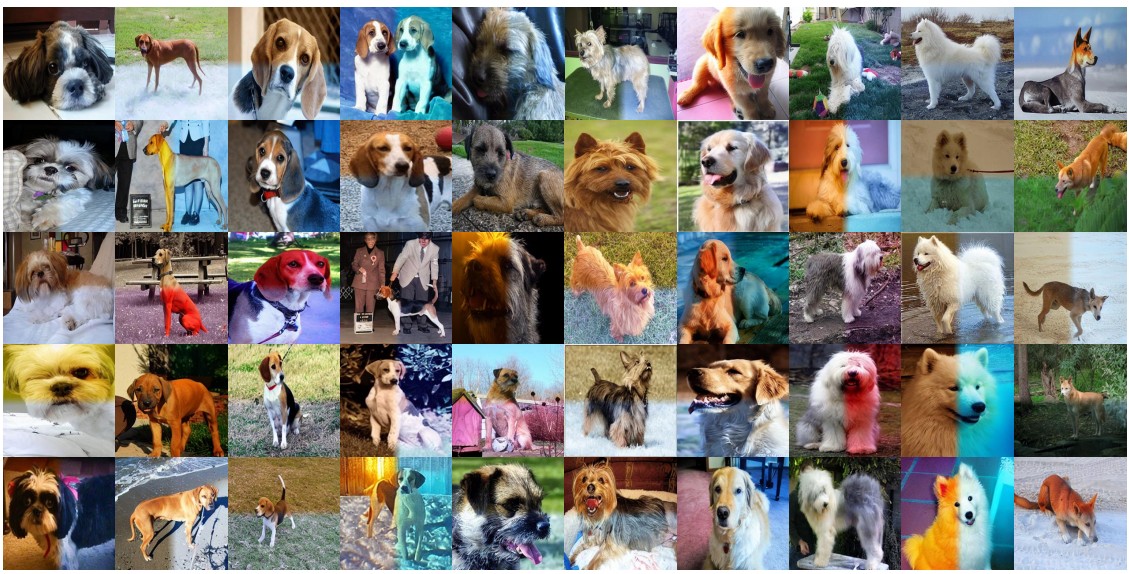

Figure 14: More visualizations selected from the condensed ImageWoof.

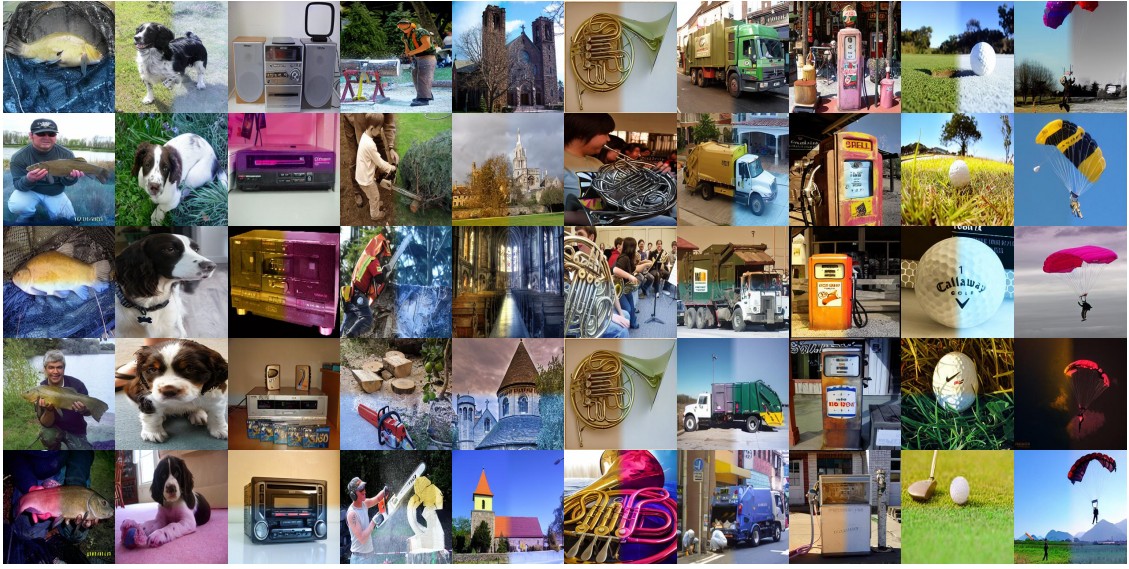

Figure 15: More visualizations selected from the condensed ImageNette.

