# OpenReview forum: "Dataset Condensation with Color Compensation"
_TMLR — Accepted by TMLR_

### Review · Reviewer_ARJN · 2025-08-19

**Summary Of Contributions:**

This paper introduces ​​DC3​​, a novel dataset condensation method that addresses two key bottlenecks in existing methods: ​​color homogenization​​ and inefficient compression. Specifically, DC3 first proposes a diffusion-based mechanism to enhance color diversity in condensed images by leveraging hue prompts. Then, DC3 replaces random bin sampling in Dataset Quantization (DQ) with a clustering-guided strategy to preserve critical samples. Experiments demonstrate robustness of proposed method across CNNs and Transformers.

**Audience:**

Yes

**Audience Explanation:**

The proposed method alleviate color homogenization​​ and inefficient compression problems existed in dataset condensation.

**Broader Impact Concerns:**

No ethical concerns.

**Claims And Evidence:**

Yes

**Claims Explanation:**

Experiments demonstrate the effectiveness of proposed method.

**Requested Changes:**

I suggest authors to conduct experiments on ImageNet to verify the effectiveness of DC3, which will make experiments more convincing.

---

> ### Author Response · Authors · 2025-09-08
> **Response to the requested changes by Reviewer ARJN**
>
> We sincerely thank the reviewers for their valuable comments and insightful suggestions! We have carefully addressed your concerns in the revised paper as detailed below.
>
> 1. Evaluation on ImageNet-1K:
> - The results of DC3 evaluation on ImageNet-1K are reported in Tab1. DC3 performs the best Top-1 Accuracy across all IPC settings.
> - We also provided comparison results on some challenging domain-specific ImageNet subsets such as ImageNette and ImageWoof in Tab2. DC3 has better performance than other dataset condensation methods, demonstrating consistent performance across varying task complexities.

---

> > ### Comment · Reviewer_ARJN · 2025-09-22
> > **Requested changes**
> >
> > Thank you for your response. In fact, I hope authors can provide experiments on ImageNet-21K rather than only using ImageNet-1K.

---

> > > ### Author Response · Authors · 2025-09-23
> > > **Response to the Comment by Reviewer ARJN**
> > >
> > > We sincerely thank you for your response and valuable advice!
> > >
> > > 1. Evaluation on ImageNet-21K:
> > > - ImageNet-21K really reflects the powerful advantages of the DD method. However, distilling it will take a very long time despite the efficient method. The experimental results will not perhaps be available to you before the end of the rebuttal phase.
> > > - Undoubtedly, we will take your consideration into account and carry out the experiments on 21K datasets. We will release the results as soon as we receive them and update our paper accordingly.

---

### Review · Reviewer_YKTQ · 2025-08-27

**Summary Of Contributions:**

This paper studies the trade-off between performance and fidelity in dataset condensation. Unlike prior works that primarily rely on quantization or distillation, this work highlights the often-overlooked color issue. To address this, the authors introduce a diffusion-based color compensation strategy (DC3) along with a submodular sampling approach. The method is intuitive and demonstrates superior performance on both traditional classification tasks and diffusion-based image generation.

**Audience:**

Yes

**Audience Explanation:**

I believe this work would be of interest to the TMLR community. Dataset condensation and distillation are active research areas in machine learning, with direct implications for data-efficient training, memory efficiency, and the scalability of large models. By focusing on the overlooked issue of color distribution—a non-trivial aspect in both classification and generative settings—the paper contributes a novel perspective that could inspire follow-up research.

Specifically, researchers in areas such as efficient dataset design, generative modeling, and data augmentation would find value in the proposed diffusion-based color compensation strategy. The demonstration that the method generalizes across architectures and also applies to diffusion generative models broadens its potential audience. Even if some details (e.g., qualitative fidelity of generated images) remain debatable, the methodological direction and empirical findings are relevant to with TMLR’s audience.

**Broader Impact Concerns:**

No concerns on the ethical implications.

**Claims And Evidence:**

Yes

**Claims Explanation:**

The paper provides a substantial amount of experimental evidence, including cross-architecture evaluations (ResNet, MobileNet, Swin-Transformer), comprehensive ablation studies on both the color compensation strategy and submodular sampling, and demonstrations on both classification and generative tasks. These results strongly support the claim that the proposed method improves upon traditional dataset condensation methods in terms of accuracy and generalization.

However, some claims are not as clearly substantiated. For example, the KDE curve analysis (Fig. 2) is difficult to reconcile with the stated claim that the distribution is “closer to the real dataset,” since the density curves diverge notably in marginal and central bins. Similarly, certain qualitative results—such as the synthetic images in Figs. 12–15 that appear visually unnatural, and the minimal visual differences in Fig. 11—make it less clear whether fidelity is truly preserved. These issues do not invalidate the contributions but do weaken the strength of some claims, particularly those concerning dataset fidelity and distributional alignment.

Overall, the evidence is convincing in demonstrating the method’s effectiveness (e.g., performance gains and efficiency), but somewhat less convincing in proving its fidelity to the original data distribution.

**Requested Changes:**

Major

- The KDE curve in Figure 2(d) is somewhat confusing. The density distribution appears significantly higher than the original data in the marginal bins, but lower in the central bins. While DC3 clearly avoids the uniform distribution issue seen in traditional DC methods, the resulting distribution also seems noticeably different from the original dataset. This raises doubts about the claim that “the KDE curve is closer to the real dataset than other methods.” In addition, Channel B in Figure 2(d) does not seem consistent with the other subfigures. Please clarify or correct me if I have misunderstood.

- Regarding the IPC concept, it is unclear whether the proposed color compensation strategy effectively increases the number of samples per class after diffusion-based enhancement. Clarifying this relationship would help position the method more precisely within the dataset condensation framework.

- The ablation study in Table 8 is also somewhat confusing. Since the focus is on sampling strategies, I would expect to see a more direct comparison, such as DC3 (with clustering) versus DC3 (with random sampling), to better isolate the benefit of clustering.

- For Figure 11, the differences between the first row (fine-tuned with the original data) and the second row (fine-tuned with the mixed data) are not visually obvious. To provide a more complete comparison, it would be helpful to include an additional row showing DiT results without fine-tuning, consistent with the setup in Figure 8.

- Finally, the condensed images shown in Figures 12–15 (from CIFAR-10/100, ImageWoof, and ImageNette) often appear unusual, with images seemingly split into two hues. Is this an expected artifact of the diffusion-based color compensation strategy? If so, further explanation is needed regarding why these results occur and how such images can still benefit generative model training.

Minor

- Switch the positions of Figures 5 and 6 for better flow.

---

> ### Author Response · Authors · 2025-09-08
> **Response to the requested changes by Reviewer YKTQ**
>
> We sincerely thank the reviewers for their valuable comments and insightful suggestions! We have carefully addressed your concerns in the revised paper as detailed below.
>
> 1. Explaination of the KDE curve:
> - If we visualize the selected samples without color compensation, the KDE curve is the same as the original dataset since they constitute a subset of the real data. Then we implement color compensation based on these samples.
> - The claim that the KDE curve of DC3 is "closer to the true color distribution" refers to relative proximity compared with traditional pixel-level optimization methods, rather than implying absolute similarity to the color distribution of original data.
> - The significant marginal bins stem from the color compensation, which has been proven to be beneficial for condensation.
> - Channel B in Figure 2(d) does not seem consistent with the other subfigures because the range of the ylim (0,2) is different from the other plots (0, 1.25).
>
> 2. Explaination of the IPC used in DC3:
> - The concept of IPC in DC3 aligns with definitions found in other dataset condensation works. The final samples used for counting IPC are obtained after Submodular Sampling and Color Compensation. For instance, Tab9 reports experiments based on IPC-50, indicating that 50 samples within each class were first selected via Submodular Sampling, followed by Color Compensation, resulting in a final set of 50 samples for evaluation. It is worth noting that the Crop-and-Stitch technique does not alter the IPC sizes.
>
> 3. Ablation study on Submodular sampling:
> - We relist the tab8 for a better understanding of the submodular sampling ablation protocol.
> - The upper section of the table omits the color compensation, and clustering-based sampling is applied only within DC3, where DQ uses random sampling.
> - The lower section of the table modifies only the sampling strategy while keeping the rest of the pipeline the same, including color compensation.
> - You can also refer to the 1st (DC3 (with random sampling)) and 2nd (DC3 (with clustering)) rows of tab10 for the results of how the clustering-based sampling strategy benefits the pipeline.
>
> 4. Visualization from Fine-tuned DiT:
> - We have included an additional row (the first row in Fig11) showing DiT results without fine-tuning, consistent with the setup in Fig8.
>
> 5. Explaination of the Crop-and-Stitch method:
> - Effectiveness: The Crop-and-Stitch method is used by DC3 to further enhance the color diversity of distilled images. We first validated the effectiveness of the Crop-and-Stitch method through the experimental results shown in Figure 7. Table 14 in Appendix B further shows the experimental results of different Crop-and-Stitch methods. The results demonstrate that half-merge is the optimal strategy for Crop-and-Stitch in DC3.
> - The reason for enhancing the generation ability of diffusion models: Since Crop-and-Stitch does not damage the semantic information in the image, and Chen et al.[1] has demonstrated that diffusion models are robust classifiers, we believe that the distilled images obtained using the Crop-and-Stitch method can improve the performance of the generative model since it can provide more information about colors without semantic collapse.
>
> 6. The readability of Fig5 and Fig6:
> - We have adjusted the position of these two figures for better readability.
>
> [1] Huanran Chen, Yinpeng Dong, Shitong Shao, Zhongkai Hao, Xiao Yang, Hang Su, and Jun Zhu. 2025. Diffusion models are certifiably robust classifiers. In Proceedings of the 38th International Conference on Neural Information Processing Systems (NIPS '24), Vol. 37. 50062–50097.

---

> > ### Comment · Reviewer_YKTQ · 2025-09-25
> > **Thanks for the rebuttal**
> >
> > Most of my earlier concerns have been addressed. The additional explanations and ablation studies indeed make the paper more comprehensive and easier to follow. I still have a few remaining questions:
> >
> > - The paper claims that the significant marginal bins arising from color compensation are beneficial for condensation. Could the authors provide supporting evidence for this claim, or point me to the relevant section if I missed it?
> >
> > - How exactly does the condensed data contribute to finetuning generative models? In Figure 11, the results appear quite similar across different approaches. Additionally, while the Crop-and-Stitch strategy preserves semantic information, it inevitably disrupts local visual coherence. I remain unconvinced about how such an operation could positively influence the training of generative models.

---

> ### Author Response · Authors · 2025-09-25
> **Response to the additional questions by Reviewer YKTQ**
>
> We sincerely thank you for your response and valuable advices! We have carefully addressed your additional questions as detailed below.
>
> 1. Explanation of the Color Compensation
> - We should make it clear that Color Compensation are benificial for condensation because of the **color distribution alignment**, rather than the significant marginal bins.
> - The significant marginal bins are just the visual representation of color compensation, the success of color compensation is due to the fact that the color distribution obtained by it (after submodular sampling) is closer to the real data, compared with the matching methods that provide the uniform color distribution.
> - In summary, the final color distribution provided by matching methods follows the uniform distribution. DC3 observed this phenomenon and provided a color distribution closer to the real data despite the significant marginal bins.
>
> 2. Explanation of fine-tuning the generative model
> - Visually, the results in Fig11 are similar, however, the quantitative results are different.
> - The FID-IS results in Fig8 **have already** demonstrated the effectiveness of DC3-finetuned diffusion models.
> - We will carry out an experiment about finetuning the diffusion models with DC3-condensed data, which only uses a single cool or warm hue prompt. It takes time to collect the results (including finetuning, generation, and evaluation), thus, we will release them once we receive them and update our paper accordingly.

---

### Review · Reviewer_JoLv · 2025-08-28

**Summary Of Contributions:**

## Summary
This paper prpoposes a new Dataset Distillation method: DC3 that makes use of Dataset Quantization (DQ) and augmentation via a diffusion model to condense a dataset into a smaller one. Previous approaches that are based on selecting samples from the dataset (basically DQ and Coreset Selection methods) have mostly been proposed for a scenario where a percentage of the dataset is retained (at least 10%). DC3 modifies the basic DQ approach primarily by doing 2 things: i) using clustering for forming bins in DQ and ii) augmenting the selected samples using diffusion models. The paper demonstrates that the proposed pipeline outperforms the baselines in low IPC (Images Per Class) settings and generalises across architectures.

### Strengths
- The pipeline is simple and modular.
- Pretrained diffusion models are used in a novel manner to augment the selected subset of samples.
- The selected subset of samples, while being small, generalises across architectures.

### Weaknesses
- The standard deviation and details around training models using the condensed datasets are missing.
- Some sections/figures need more clarification and details.
- Depending on the details that are missing, more ablations might be required.

**Audience:**

Yes

**Audience Explanation:**

Dataset Distillation has gained popularity over the years (https://github.com/Guang000/Awesome-Dataset-Distillation), and this manuscript explores a less explored direction of selecting samples from the dataset to condense it while keeping the total number of selected samples low. I think the dataset distillation community will find this paper intriguing, not only because it advances SOTA but also because of the novel pipeline and for showing that one can achieve high accuracy even with sample selection (Dataset Quantisation).

**Broader Impact Concerns:**

There are no concerns on the ethical implications of this work.

**Claims And Evidence:**

Yes

**Claims Explanation:**

My major concerns/questions/requests are the following:

1. **Standard deviation across random seeds** Please report standard deviations for all numeric results across multiple random seeds. As of now, it's unclear whether the reported results are for one best-performing model trained on the respective condensed dataset or an average across runs. Please also report how the models are trained and how the best checkpoint (for reporting numbers on the test set) is selected for every run and every method.

2. **Page 10 — “Compensation vs. Augmentation” protocol.** When comparing Colour Compensation to Augmentation, do you only replace the Colour Compensation module while keeping the rest of the DC3 pipeline unchanged? If not, then that would be an ablation that is required.

3. **Page 11 — submodular sampling ablation protocol.** For the submodular sampling variant, please state explicitly whether (a) Colour Compensation is omitted and clustering is applied only within Dataset Quantisation (DQ), or (b) Colour Compensation remains and only the sampling strategy is replaced. Another ablation required would be the one that complements the one in point 2: modify only the sample selection strategy while keeping the rest of the pipeline the same.

**Requested Changes:**

### Critical for securing my recommendation for acceptance

1. **Fig.5** I did not fully understand Fig. 5. Does each subplot show a different class (label)? And what you're trying to show is that the clusters are spread out?

2. **Fig.6** I did not understand Fig. 6. What do the different columns signify? And are you trying to show that samples should be as far apart as possible, but they are not? You have pointed out some of the subplots in the main text but the overall idea behind the figure is unclear. Also, in the middle, all the submodular gains are very spread out and do not seem to follow a pattern so its unclear how the colouring scheme is helping.

3. **Page 11 / Table 9 — what does “DC3” denote, and what is the IPC here?** In Table 9, does the label “DC3” refer to the full DC3 pipeline or only a subset of components? Also, please report the images-per-class (IPC) used for each row in the table caption.

### Not critical for securing my recommendation for acceptance, but should ideally be fixed

1. **Abstract — clarify “inefficiency condensation”.** Do you mean *“inefficiency in condensation”* or *“inefficient condensation”* or something else?

2. **Abstract — expand abbreviations on first use.** Please write out the full form of every abbreviation the first time it appears in the manuscript (for example, *Frechet Inception Distance (FID)* in the abstract).

3. **Page 3, line 1 — unclear sentence.** I did not understand the sentence *“We propose DC3 that utilizes the compression ability as long as the generalization from different dataset condensation methods.”*. Please reword this sentence for clarity.

4. **Page 4, §3.1 — missing citation for CLIP.** CLIP is referenced in the notation for latent diffusion conditioning but lacks a formal citation. Please add the CLIP citation where it is first mentioned.

5. **Page 4, 2nd last line — clarify “except for the neutral colors”.** I did not understand what you meant by the phrase *“i.e., except for the neutral colors”* in the context of the whole sentence.

6. **Page 5, last paragraph — define “optimal sample”.** The manuscript refers to an *“optimal sample”* without a formal definition. Please define this term precisely (for example, as the sample maximizing a specified submodular gain or distance metric).

7. **Missing short description of RDED.** RDED is used as a baseline and as part of the narrative in the experiments section, but no short, self-contained description is given.

8. **Section 4.1 — use of soft labels and ResNet18.** It is unclear whether soft labels are applied only to RDED or uniformly to all methods that require soft labels. Same for ResNet18.

9. **Table 4 — define “seen” and “unseen” architectures.** What all architectures are included in *“seen”* and *“unseen”* sets?

10. **Page 12, §4.5 — expand and cite FID-IS.** The term *“FID-IS”* is used without definition or citation. Please expand the abbreviations on first use (e.g., *Frechet Inception Distance (FID)*, *Inception Score (IS)*) and cite the original references. Note: you may want to cite FID in the abstract and then simply use it here directly, but IS does need a citation and expansion.

---

> ### Author Response · Authors · 2025-09-08
> **Response to the requested changes by Reviewer JoLv**
>
> We sincerely thank the reviewers for their valuable comments and insightful suggestions! We have carefully addressed your concerns in the revised paper as detailed below.
>
> 1. The missing standard deviation and details:
> - All of the DC3 results we reported are an average across 3 runs.
> - We select the best checkpoints based on their performance (Accuracy) on the validation set for each run.
> - We have added the standard deviation for DC3 and other results we reproduced. The corresponding details are added in the revised paper.
>
> 2. Compensation vs. Augmentation protocol:
> - Yes, the only variable in this experiment is the color method, the other components in the pipeline are the same.
>
> 3. Submodular sampling protocol:
> - We relist the tab8 for a better understanding of the submodular sampling ablation protocol.
> - The upper section of the table omits the color compensation, and clustering-based sampling is applied only within DC3, where DQ uses random sampling.
> - The lower section of the table modifies only the sampling strategy while keeping the rest of the pipeline the same, including color compensation.
>
> 4. Explaination of Fig5 & Fig6:
> - Fig5: Each subplot in Fig5 shows the samples from one selected label. These samples are then divided into different bins by the clustering algorithm.
> - Fig6: The middle row indicates that different samples with different Submodular Gain are randomly distributed in the dataset. The original intention of using Submodular Gain to divide dataset is to improve the diversity (make selected samples more spread out). However, if you select samples randomly in the divided bins, the quantized samples lack diversity as depicted in the top-center and bottom-left subplots (they are close to each other). Different columns mean different classes.
> - Explaination: DQ divide samples into bins according to the Submodular Gain, which will result in a lack of diversity (Fig6) and thus affect accuracy. DC3 uses clustering algorithms to divide bins, followed by Submodular Sampling, which ensures the diversity (because of the clustering) and representativeness (because of the submodular sampling) as shown in Fig5.
>
> 5. Details for Tab9：
> - DC3 refer to the full pipeline since there is no special annotation like a dagger. The IPC is 50 as we reported in table caption.
>
> 6. Changes for better readability
> - Abstract: inefficiency condensation -> inefficiency in condensation
> - Abstract: We provide full form of FID and other abbreviations for the first time they appear in our manuscript.
> - Contribution 1: revised as "According to the dataset condensation triangle, we propose DC3 that utilizes the compression ability of dataset distillation along with the generalization ability of dataset quantization."
> - Sec3.1: Citation of CLIP is added.
> - Sec3.2: i.e., except for the neutral colors -> ~~i.e.,~~ except for the neutral hues (e.g., black, white, and grey)
> - Sec3.3: definition of "optimal sample": the sample with the highest submodular gain.
> - Related works: Added description about RDED.
> - Sec4.1: The soft label protocol is used for all methods that require it. The Teacher model is fixed as ResNet-18.
> - Tab4: We added the description about "seen" and "unseen" architectures.
> - Sec4.5: We provide the full form of FID-IS and added the citations.

---

> ### Comment · Reviewer_JoLv · 2025-09-14
> **Additional questions (clarifications)**
>
> Thank you for answering the questions and addressing the concerns.
>
> 1. **Fig. 5 (now Fig. 6):** The sentence in the main text for this fig. says, *"As shown in fig. 6, data bins, also known as clusters, are dispersed throughout the feature space as expected."* and the figure's caption reads *"Submodular sampling selects the representative samples to preserve maximum semantic integrity and feature diversity."* 2 questions: what do you mean by "semantic integrity" and what's the main reason for having this figure in the manuscript? Is it show that the clusters are spread out so when you select a sample from each cluster you're selecting diverse samples? Also, please modify the caption to clearly indicate that each subplot shows clustering for a different label (maybe list the labels if space allows).
>
> 2. **Fig. 6 (now Fig. 5):** If I understand the figure correctly, you're trying to show: **a.** In the middle row, the per-sample gain values are scattered across the feature cloud (i.e. high and low gains are mixed). This means "high gain" does not necessarily mark a separate region of the feature space; high-gain points can be near each other or near low-gain samples. **b.** Because DQ builds bins by repeatedly choosing top-gain items, bins can end up with samples that are not well-spread spatially. After bins are formed, DQ then samples uniformly within bins. However, if many diverse samples end up in the same bin (because their gains are high), uniform sampling will typically pick only one and discard the rest, thereby losing diversity. However, if you examine the first row, the 2nd and 3rd columns appear to be opposites. Are you suggesting that it can be random, in the sense that it sometimes works in our favour and sometimes not? Perhaps you could also illustrate how DC3 selects the samples here, in contrast to the DQ? Also, please specify what each column indicates in the figure's caption.
>
> 3. Thank you for specifying the "seen" and "unseen" architectures in Table 4. But please list the exact architectures used (both convolution-based and transformer-based architectures)...perhaps in the appendix if there's not enough space in the main text.
>
> 4. Regarding the validation set, does it come from the distilled dataset or is it a separate subset that comes from the original training data? Also, how big is the validation set for different IPC settings?

---

> ### Author Response · Authors · 2025-09-17
> **Response to the additional questions by Reviewer JoLv**
>
> We sincerely thank you for your response and valuable advice! We have carefully addressed your additional questions in the revised paper as detailed below.
>
> 1. Explaination of the t-SNE plots of DC3 and DQ (Fig5 & Fig6):
> - The semantic integrity mainly refers to DC3 not destroying in pixel space. We have removed the expression of semantic integrity to avoid confusion.
> - In the DQ paper, the authors believe that the selected samples should satisfy 1) diversity (spread out in the t-SNE plot) and 2) high submodular gain.
> - However, the real situation in DQ is shown in Fig5: the selected sample may not be diverse (bottom-left subplot) or have low submodular gain (bottom-right subplot). The diversity and high submodular gain sample (top-right subplot, the sample with the 2nd highest submodular gain) is dropped due to the random sampling used in DQ.
> - DC3 first implements clustering, then selects samples within each bin according to the submodular gain rather than random sampling.
> - Consequently, as shown in Fig6, the samples selected by DC3 will spread out on different bins, which guarantees both their diversity and high submodular gain. The purpose of Fig6 is to inform readers that the selected samples are indeed separated.
> - In Fig5 and Fig6, each column represents different categories. We have added the description in the revised version.
>
> 2. The "seen" and "unseen" architectures:
> - We added the specific architectures' names in the revised version.
>
> 3. Explanation of the validation set:
> - The validation sets come from the original datasets, which means they are real and not the distilled (synthetic) samples.
> - For example, we distilled the ImageNet-1K training set with DC3 and used it to train a network like ResNet-18. Then we apply the real 'val' set of ImageNet-1K to implement the evaluation phase and obtain the reported results.
> - The size of the validation set is independent of IPC.

---

> > ### Comment · Reviewer_JoLv · 2025-09-19
> > **Usage of validation sets**
> >
> > 1. Are the validation sets for datasets separate from the test sets so that you have 3 sets: train, val and test? Or are the validation sets the same as the test sets?
> >
> > 2. Were the hyperparameters for your method (and baseline methods, if required) primarily tuned on the validation sets, at least for the datasets where the validation set is sufficiently large?

---

> ### Author Response · Authors · 2025-09-20
> **Response to the additional questions by Reviewer JoLv**
>
> We sincerely thank you for your response! We have carefully addressed your additional questions below.
>
> - Using ImageNet (from ImageNet Large Scale Visual Recognition Challenge, ILSVRC) as an example, the dataset is officially partitioned into three subsets: a training set, a validation set, and a test set. While the training and validation sets include labels, the original test set lacks ground truth annotations. Therefore, we evaluate the neural network trained on the distillation dataset using the labeled validation set and report the corresponding accuracy.
> - The ImageNet subsets have the same dataset structure as 1K.  Thus, the test sets are still unlabeled, and we can only use the validation set for evaluation.
> - So yes, the hyperparameters for DC3 and baseline methods are primarily tuned on the validation sets, which aligns with the standard evaluation protocol.

---

> > ### Comment · Reviewer_JoLv · 2025-09-20
> >
> > 1. For IN1K, it is common to use the official validation set as the test set but it's also common to partition the official train set into train and val sets for hyperparameter tuning and checkpoint selection.
> >
> > 2. But what about the other datasets, do you use 3 separate sets: train, val and test or just 2 like IN1K?
> >
> > 3. Do, you train, tune (hyperparameters) and test all baseline methods using the same protocol?

---

> > > ### Author Response · Authors · 2025-09-20
> > > **Response to the additional questions by Reviewer JoLv**
> > >
> > > We sincerely thank you for your response! We have carefully addressed your additional questions below.
> > >
> > > 1. Evaluation on other datasets:
> > > - Overall, our evaluation follows these principles:
> > >   - For the IN1K subsets, we use the same evaluation protocol as IN1K
> > >   - For CIFAR-10/100, we use the latter protocol as you said, i.e., partitioning the train set into train and val sets during the hyperparameter tuning and checkpoint selection, and use the original val set to report the results.
> > >
> > > 2. Evaluation protocol:
> > > - We use the above principles for the results of DC3 and the methods we reproduced.
> > > - Other results are collected from their papers directly.

---

> > > > ### Comment · Reviewer_JoLv · 2025-09-20
> > > >
> > > > Thank you, I think that answers all the questions I had.

---

### Review · Reviewer_hcBN · 2025-09-08

**Summary Of Contributions:**

This manuscript introduces DC3, a data augmentation technique, applied to the task of data condensation. This framework combines clustering-based quantization with submodular sampling and color compensation using a diffusion model. The main claims are that other data condensation techniques generally suffer from color homogenization, and DC3 solves this problem by color augmentation using the Latent Diffusion Model (LDM). However, it is not specified which model is used. The model is presented as efficient and training-free. Experiments performed on CIFAR, ImageNet, ImageWoof, and Tiny-ImageNet show that DC3 outperforms state-of-the-art methods for data condensation.

**Additional Comments:**

Clarity: Figures and explanations (e.g., Algorithm 1, submodular gain) could be made clearer; currently, the pipeline is somewhat difficult to follow.

Fairness of baselines: Since DC3 augments its samples via diffusion, other baselines should be allowed similar augmentation to ensure fairness.

Applicability: The method is well-suited for low-data augmentation scenarios (e.g., medical imaging, rare-event domains), but less compelling as a dataset condensation method where the main motivation is computational efficiency.

**Audience:**

Yes

**Audience Explanation:**

Although the manuscript makes several omissions and is not exhaustively tested, it presents very good results compared to other models; however, this should be taken with caution.

**Claims And Evidence:**

No

**Claims Explanation:**

1 Ambiguity of diffusion model

The paper only mentions using a Latent Diffusion Model (LDM) but does not explicitly state which model or checkpoint is used. While Section 4.5 (experiments) mentions that two diffusion models (Stable Diffusion and Diffusion Transformers) are trained, Section 3 (method) fails to mention which model was used to generate new images for color variation. While the citation to Rombach et al. 2022 suggests Stable Diffusion, this should have been stated clearly for reproducibility.

2 Effective dataset size inflation

Reported IPC budgets (e.g., IPC-10 or IPC-50) are misleading. In practice, DC3 multiplies the dataset size by generating more color-compensated variants per selected sample. For example, at IPC-10 on ImageWoof, instead of training with 10 images, the final dataset contains 100 images due to warm/cool prompt variations (Table 11). This undermines fair comparison to methods that strictly adhere to IPC budgets.

3 Unaccounted computational cost

The method is described as comutational efficient, yet generating hundreds of diffusion-augmented samples per class is computationally heavy. The paper omits reporting inference cost, runtime, or GPU resources required for augmentation, which are critical for evaluating practicality. The main objective of dataset condensation is that we have a very large database, and it is necessary to reduce it intelligently in order to reduce the training time and reduce the computational cost.

4 Potential information leakage from diffusion model

The diffusion backbone (trained on large external datasets) likely introduces external knowledge into the augmented data. This could explain strong cross-architecture generalization, but it blurs the line between dataset compression and data augmentation with external priors. The paper does not acknowledge or analyze this risk.

5 Lack of theoretical grounding
The paper provides no formal justification for why color diversity should be so crucial to generalization. Claims about submodular sampling and color compensation remain empirical observations (KDR curves) without deeper theoretical analisis. Correlation does not mean causation.

6 Evaluation scope

Comparisons to recent Dataset Quantization (DQ) methods (ADQ, DQAS, 2024–25) are missing. These methods are natural baselines given DC3’s use of clustering and quantization. The method is also not tested at moderate keep-ratios (10–30%), where DQ methods excel, making it unclear whether DC3’s advantage holds beyond extreme IPC settings.

**Requested Changes:**

no

---

> ### Author Response · Authors · 2025-09-09
> **Response to the requested changes by Reviewer hcBN**
>
> We sincerely thank the reviewers for their valuable comments and insightful suggestions! We have carefully addressed your concerns in the revised paper as detailed below.
>
> 1. Ambiguity of diffusion model:
> - For LDM, we use Stable Diffusion-V1.5 as the generative model and fine-tuning model. For DiT, we use DiT-XL/2-256.
> - The corresponding details are added in the revised paper.
>
> 2. Effective dataset size inflation:
> - In our experiment, we randomly selected one prompt from both cool and warm hues for color compensation. Then we crop-and-stitch the generated image to create one single distilled image.
> - We repeat the above process 10 times to obtain 10 distilled samples and then use them to implement the downstream classification tasks.
> - Thus, the dataset size is not inflated, and the comparison is fair enough.
>
> 3. Unaccounted computational cost:
> - According to response 2, we need to clarify again that our method does not extensively generate hundreds of diffusion-enhanced images at once.
> - We have provided the comparison of computation time and GPU memory cost in Appendix F.
>
> 4. Potential information leakage:
> - Unlike other DC methods that employ diffusion models for dataset condensation, our approach does not generate any images with novel content, such as new objects or backgrounds, using diffusion models. Instead, diffusion models are solely used for adjusting the color attributes of original samples, preserving semantic content and structural shapes while modifying only color information.
> - The cross-architecture generalization capability of DC3 does not stem from the use of diffusion models, but rather, similar to DQ, from the **selection operation** of key samples directly from raw data. This operation ensures alignment with the original data distribution without introducing synthetically generated samples.
> - As illustrated in Figures 3 and 4, the distilled images remain semantically identical to the original data.
>
> 5. Lack of theoretical grounding:
> - We strongly argue that our paper does address the causal impact of color diversity on generalization through a series of ablation studies, which move significantly beyond mere observational correlation.
>     - Systematic Manipulation of Color Diversity: As shown in Tab6, we systematically manipulated the level of color diversity through different prompt settings. The results clearly demonstrate that reducing the diversity of color prompts directly led to a measurable decrease in generalization performance of 3.3%\~7.5% (CIFAR-10, IPC-10) and 3.9%\~5.6% (CIFAR-100, IPC-10).
>     - Direct Impact of Color Compensation: In Fig7, we directly compare the effect of using no-hue, single hue vs mixed hue, showing that models trained with mixed hue distilled samples outperform those trained without compensation, the gap ranges from 9.5%~27%.
>     - Ablation of DC3 Pipeline: Tab10 presents a crucial ablation study on our DC3 pipeline. It reveals significant performance improvements derived from both the color compensation and the submodular sampling.
> - The results of these ablation studies clearly demonstrate a direct and causal link: a reduction in color diversity consistently leads to a significant degradation in generalization performance. Conversely, our proposed methods for compensating color diversity (e.g., the color compensation mechanism), when ablated, show a detrimental effect on model performance.
> - These controlled experiments, where we actively intervened with and manipulated the color diversity aspect, provide strong evidence of a causal relationship, rather than merely observational correlation.
> - Our color compensation mechanism directly addresses potential biases in the uniform distilled data color distribution, a common issue that can hinder generalization. By actively balancing color representation, we prevent the model from overfitting to dominant color patterns.
> - We acknowledge that formalizing these insights into a complete mathematical theory is a complex task and is indeed a valuable direction for future work.
>
> 6. Evaluation scope:
> - The ADQ and DQAS you mentioned mainly focus on distilling the small-scale and low-resolution (32x32) datasets such as CIFAR-10 and CIFAR-100.
> - The lowest IPC reported in ADQ is 500, while the highest IPC reported by most DC methods is 100. Since the experimental setups and the IPCs they reported are quite different from other DC methods, we did not include them in the comparative experiments.
> - The following table compares the Top-1 Accuracy of DQAS (reported from their paper) and DC3 under IPC-50:
>
> |  Method   | CIFAR-10  | CIFAR-100 | Tiny-ImageNet |
> |  ----  | ----  | ----  | ----  |
> |  DQAS  | $52.3$  |$ 52.6$ | $53.4$ |
> | DC3  | $80.9\pm0.5$ | $64.2\pm0.5$ | $59.4\pm0.7$ |
> - However, the comparison is unfair because the surrogate evaluation network is different. Thus, we did not report these results.

---

### Author Response · Authors · 2025-09-08
**Official Comment from Authors**

We sincerely thank the reviewers for their valuable comments and insightful suggestions! We have addressed the reviewer comments, suggestions, and clarifications. The addressed changes are highlighted in blue.

---

### Decision · Action_Editor_xyMa · 2025-10-19

**Recommendation:** Accept as is

**Audience:**

Yes

**Audience Explanation:**

The TMLR audience, particularly those interested in efficient training, data-centric AI, and model compression, would find the findings valuable.

**Claims And Evidence:**

Yes

**Claims Explanation:**

The claims made in the submission are supported by accurate, convincing, and clear evidence. The proposed DC3 method effectively addresses the two main limitations in existing dataset condensation approaches—color homogenization and inefficient compression. The authors present comprehensive experiments across both CNN and Transformer architectures, and provide detailed ablation studies that validate the contributions of each component.

---

> ### Author Response · Authors · 2025-10-22
>
> We sincerely thank the Action Editor and all reviewers for their time and valuable feedback throughout the review process. The quality of our paper has significantly improved thanks to their constructive comments and suggestions. We will carefully incorporate the remaining feedback and aim to submit the final camera-ready version as early as possible.